# GROUP DOWNSAMPLING WITH EQUIVARIANT ANTI-ALIASING

**Md Ashiqur Rahman**
Department of Computer Science
Purdue University
rahman79@purdue.edu

**Raymond A. Yeh**
Department of Computer Science
Purdue University
rayyeh@purdue.edu

## ABSTRACT

Downsampling layers are crucial building blocks in CNN architectures, which help to increase the receptive field for learning high-level features and reduce the amount of memory/computation in the model. In this work, we study the generalization of the uniform downsampling layer for group equivariant architectures, e.g., $G$-CNNs. That is, we aim to downsample signals (feature maps) on general finite groups *with* anti-aliasing. This involves the following: **(a)** Given a finite group and a downsampling rate, we present an algorithm to form a suitable choice of subgroup. **(b)** Given a group and a subgroup, we study the notion of bandlimited-ness and propose how to perform anti-aliasing. Notably, our method generalizes the notion of downsampling based on classical sampling theory. When the signal is on a cyclic group, *i.e.*, periodic, our method recovers the standard downsampling of an ideal low-pass filter followed by a subsampling operation. Finally, we conducted experiments on image classification tasks demonstrating that the proposed downsampling operation improves accuracy, better preserves equivariance, and reduces model size when incorporated into $G$-equivariant networks.

## 1 INTRODUCTION

Computer vision models, *e.g.*, ConvNets or ViTs, consist of striding and pooling layers used for downsampling a feature map (He et al., 2016; Liu et al., 2022; Dosovitskiy et al., 2021; Liu et al., 2021b; Wu et al., 2021; Yang et al., 2024). These subsampling layers play a crucial role in learning the spatial hierarchy of features, building in translation invariance, and reduction in computation (Zhang et al., 2023). Concepts from signal processing (Vetterli et al., 2014), *e.g.*, bandlimited-ness and anti-aliasing, have also been introduced to design better downsampling (anti-aliasing followed by subsampling) operations (Zhang, 2019; Zou et al., 2020; Vasconcelos et al., 2021).

Given additional prior knowledge, group equivariant ConvNets and Transformers have been proposed to incorporate additional structure into the models (Cohen & Welling, 2016; Tai et al., 2019; Romero & Cordonnier, 2021; Rojas-Gomez et al., 2022; 2024; Xu et al., 2023). These models have guarantees that the output is transformed predictably when the input is transformed. A canonical example is shift-equivariance in image segmentation, where the output mask is shifted accordingly when the input image is shifted.

Interestingly, subsampling layers are not as common in group equivariant architectures. Most models only subsample over the translation group. One limitation is that existing subsampling layers (Cohen & Welling, 2016; Xu et al., 2021) over groups *require knowing the subgroup* to downsample to. That is, there is no notion of "subsampled by a factor of two". From a practitioner's point of view, it is often unclear how to choose such a subgroup (see Appendix §A1.4 for an example). Furthermore, these subsampling layers are not designed with proper anti-aliasing, which hurts the equivariance guarantees (Gruver et al., 2023).

In this work, we propose a generalization of uniform downsampling of signals (features maps) on general finite groups with anti-aliasing. We present an algorithm to form a suitable choice of subgroup given a finite group and an integer downsampling factor. Next, we define the sampling theorem and bandlimited-ness for subgroup subsampling of signals on groups. To ensure the signal is bandlimited, we propose an anti-aliasing operation following the introduced bandlimited definition

while maintaining equivariance. We point out that our proposed algorithm and definitions intuitively generalize the notion of downsampling based on classical sampling theory.

Beyond the theoretical aspects, we conduct experiments to test the proposed downsampling operation. First, we numerically validate the proposed claims. Second, we conduct experiments on the MNIST and CIFAR-10 datasets to evaluate the performance of the proposed downsampling layer on image classification tasks over different symmetries. We show that our proposed subsampling layer selects suitable subgroups for the task of image classification, and the proposed anti-aliasing operation further improves the models' performance both in task performance and equivariance. **Our contributions:**

- We generalize the uniform subsampling operation to signals on finite groups, allowing subsampling at a desired rate, yielding signals on subgroups.
- We introduce the Subgroup Sampling Theorem and the concept of bandlimited-ness for subgroup subsampling. It guarantees the perfect reconstruction of the signal on the whole group from the subsampled signal on the subgroup.
- We propose an equivariant anti-aliasing operation to ensure the signals are bandlimited before subgroup subsampling. Empirically, we demonstrate the efficacy of the downsampling operation.

## 2 RELATED WORKS

**Downsampling layers (subsampling & anti-aliasing).** The idea of subsampling has rooted in striding and pooling as early as the seminal works of CNNs (Fukushima, 1980; LeCun et al., 1999). To downsample a high-resolution feature map to a low-resolution one, *e.g.*, by a factor of two, one can simply discard every other element in a feature map. More recently, anti-aliasing has been incorporated into deep nets, inspired by signal processing, where they propose to blur the feature map before subsampling using a low-pass filter (Zhang, 2021; Karras et al., 2021; Rahman & Yeh, 2024). Later, subsampling has also been extended to groups (Cohen & Welling, 2016; Xu et al., 2021). However, the term "every other" is ambiguous here, resulting in a definition that assigns groups to specific subgroups without adequately addressing the subsampling rate. Additionally, the anti-aliasing operation is not tailored for groups. In contrast, this work addresses these limitations by creating a theoretical foundation for subsampling by a specified factor within groups and proposing an effective anti-aliasing method that extends the sampling theorem, which we discuss next.

**The sampling theorem** is the basis of digital signal processing, which studies how to sample, interpolate, and manipulate signals sampled at different rates (Vetterli et al., 2014). The sampling theorem guarantees that bandlimited signals can be perfectly reconstructed given a high enough sampling rate. This idea has been extended to graph signals and the field of graph signal processing (Chen et al., 2015b;a). The sampling theorems for the cyclic and abelian groups have also been studied (Dodson, 2007; Faridani, 1994; McEwen et al., 2015; Napolitano & Spooner, 2001; Vaidyanathan & Kirac, 1999). These works generalize the discrete Fourier transform to discrete groups but do not consider a generalization for all finite discrete groups. Different from these works, we present a generalization of the sampling theorem for any finite groups, propose a downsampling layer, and show how the layer can be incorporated into group equivariant deep-nets.

**Equivariant deep-nets.** Incorporating equivariance into deep-nets has been found to be an effective approach to designing deep nets (Cohen & Welling, 2017; Bekkers et al., 2018; Worrall & Welling, 2019; Yeh et al., 2022) across many applications in multiple domains, *e.g.*, sets (Ravanbakhsh et al., 2017; Zaheer et al., 2017; Hartford et al., 2018; Yeh et al., 2019a; Liu et al., 2021a; Rahman et al., 2024), graphs (Maron et al., 2019; Liu et al., 2020; Yeh et al., 2019b; Morris et al., 2022; Liao & Smidt, 2023; Du et al., 2023), etc. We foresee that our proposed downsampling layer can be incorporated into this rich literature of group equivariant architectures to build more effective and efficient models.

## 3 PRELIMINARIES

We now review the necessary background and definitions. For further details, please refer to §A1.

**Downsampling of sequences.** Given a subsampling factor $R$ and a signal $\boldsymbol{x} \in \mathbb{R}^N$, the subsampling operation is defined as

$$\mathtt{Sub}_R : \mathbb{R}^N \to \mathbb{R}^{\lfloor N/R \rfloor} \ \ \forall N \in \mathbb{Z}^+, \text{ where } \mathtt{Sub}_R(\boldsymbol{x})[n] \triangleq \boldsymbol{x}[Rn]. \tag{1}$$

When we subsample a signal following Eq. (1), it can often result in a distorted signal due to aliasing. To avoid this, an anti-aliasing filter is used to remove high-frequency content, *i.e.*, to obtain a *bandlimited signal*, before subsampling. An ideal *anti-aliasing filter*, denoted as $\mathbf{h}$, is used to remove all frequency content above the Nyquist frequency (Shannon, 1949). In summary, the ideal *downsampling* can be expressed as an anti-aliasing filter followed by the subsampling operation as:

$$\mathtt{Dwn}_R(\mathbf{x})[n] = (\mathbf{x} * \mathbf{h})[Rn], \text{ where } \mathcal{F}(\mathbf{h})[i] = 1 \text{ if } i \leq f_{\mathtt{Nyquist}} \text{ else } 0. \tag{2}$$

Here, $\mathcal{F} : \mathbb{R}^n \to \mathbb{C}^n$ denotes the discrete Fourier transform.

*Remarks:* Subsampling a finite sequence involves retaining the signal at every $R$-th factor. At a glance, it is not obvious how to generalize this subsampling strategy to a finite group $G$. As $G$ is a set, there is no notion of "every $R^{\text{th}}$" element. Naively sorting the elements and applying the subsampling for sequences would also not work, *e.g.*, the subsampled set may not be a subgroup.

$G$**-Equivariance.** In deep learning, imposing equivariance on the layers is often desirable. We say a linear map (layer) $\boldsymbol{W} \in \mathbb{R}^{n' \times n}$ is equivariant with respect to a group $G$ with representation $\rho_{U} : G \to GL(U)$ and $\rho_{U'} : G \to GL(U')$ with $U \subseteq \mathbb{R}^n$ and $U' \subseteq \mathbb{R}^{n'}$ if

$$\boldsymbol{W} \rho_{U}(g)u = \rho_{U'}(g)\boldsymbol{W}x \ \ \forall g \in G, \forall u \in U. \tag{3}$$

**Generating set.** A subset $S$ of group $G$ is said to be the *generating set* if any element $g \in G$ can be expressed as a product of the elements of $S$. We use the notation $G = \langle S \rangle$ to denote that $G$ is generated by $S$ and assume identity element $e \notin S$. The set $S$ is called the *minimal generating set* when $\langle S \backslash \{s\} \rangle \neq G \ \forall s \in S$, *i.e.*, every element of $S$ is necessary to generate the group $G$. Note, a group can have more than one minimal generating set. We call the $k^{\text{th}}$ power of an element $s \in S$ *non-redundant* if $s^k$ cannot be expressed as a product of the rest of the generating elements $S \backslash \{s\}$ when $s^k \neq e$.

**Cayley graph.** To better understand the abstract structure of a group, one approach is to represent it as a graph, namely, *Cayley graph*. Given a group $G$ and its generating set $S$, a Cayley graph $\Gamma(G, S)$ consists of vertices $V$ and edges $E$. The vertices correspond to each element $g \in G$, and there exists an edge $(a, b) \in E$, if there exists an $s \in S$ such that $b = a \cdot s$. In the directed Cayley graph, edges are directed from $a$ to $b$. Any element $g \in G$ can be represented as a path on the $\Gamma(G, S)$ starting from the identity node $e$.

**Fourier transform for finite groups.** The notion of Fourier transform has also been studied on groups (Folland, 2016; Stankovic et al., 2005). For a finite group $G$, let $\hat{G}$ be the set of complex unitary *irreducible representations (complex irreps)*. We denote the dimensionality of an irrep $\varphi \in \hat{G}$ as $d_\varphi$ such that $\varphi(g) \in \mathbb{C}^{d_\varphi \times d_\varphi} \ \ \forall g \in G$. The Fourier transform of a square-integrable function $f \in L^2(G)$ is

$$\hat{f}(\varphi_i^{mn}) = \frac{1}{|G|} \sum_{g \in G} \sqrt{d_{\varphi_i}} f(g) \overline{\varphi_i^{mn}(g)} \ \ \forall \varphi_i \in \hat{G} \text{ and } 1 \leq m, n \leq d_{\varphi_i}, \tag{4}$$

where $\varphi_i^{mn}(g)$ denotes the entry at $m^{\text{th}}$ row and $n^{\text{th}}$ column for matrix $\varphi_i(g)$. Next, $\hat{f}(\varphi_i^{mn})$ denotes the *Fourier coefficient* corresponding to irrep component $\varphi_i^{mn}$. Similarly, the inverse Fourier transform on a group can be expressed as

$$f(g) = \sum_{\varphi_i \in \hat{G}} \sum_{mn \leq d_{\varphi_i}} \hat{f}(\varphi_i^{mn}) \sqrt{d_{\varphi_i}} \varphi_i^{mn}(g), \tag{5}$$

where we denote the set of orthonormal Fourier basis as $\{\sqrt{d_{\varphi_i}} \varphi_i^{mn} | \varphi_i \in \hat{G} \text{ and } m, n \leq d_{\varphi_i}\}$ following the Peter-Weyl theorem (Peter & Weyl, 1927). For *real-irreps*, the orthonormal basis set is constructed by only taking non-redundant columns of $\varphi_i$ (see Supp C of Cesa et al. (2021)).

## 4 METHOD

In this work, we consider signals $x \in \mathcal{X}_G \triangleq \{x : G \to \mathbb{R}^d\}$ to be an unconstrained real-valued function over a finite group $G$. For readability, we describe the content with $d = 1$, which can be easily generalized. A group element $g$ acts on the space $\mathcal{X}_G$ via a regular representation $\rho_{\mathcal{X}_G}$, *i.e.*, $(\rho_{\mathcal{X}_G}(g)x)(u) = x(g^{-1}u) \ \ \forall u \in G$. Our goal is to design a downsampling operator $\mathtt{Dwn}_r^G : \mathcal{X}_G \to \mathcal{X}_{G^\downarrow}$, which resamples the signal on a group $G$ to be on a subgroup $G^\downarrow \subset G$.

This involves addressing the following: **(a)** Given a group $G$ and a subsampling rate $R$, what is an appropriate subgroup $G^{\downarrow}$? **(b)** Given a group and a subgroup, what is the notion of bandlimited-ness to guide the design of anti-aliasing? We answer these questions by proposing a downsampling operation that generalizes the existing notion of subsampling (§4.1) and sampling theorem (§4.2) from sequences to finite groups.

### 4.1 UNIFORM GROUP SUBSAMPLING

A natural generalization from subsampling of sequences in Eq. (1) to signals on a group is to keep the signal on a subgroup $G^{\downarrow}$ and discard the rest:

$$x^{\downarrow} = \mathtt{Sub}_R^G(x) \text{ with } x^{\downarrow}[g'] = x[g'] \quad \forall g' \in G^{\downarrow}, \quad (6)$$

where the downsampled signal is denoted by $x^{\downarrow} : G^{\downarrow} \to \mathbb{R}$. However, it is not obvious how to obtain such $G^{\downarrow}$ and how to relate it to the rate $R$.

Given a group $G$ and a subsampling rate $R$, we propose the *uniform group subsampling*, which returns a subgroup $G^{\downarrow}$. Our subsampling algorithm intuitively generalizes from the traditional subsampling and is guaranteed to return a subgroup under mild conditions (details in Clm. 1). Our approach breaks subsampling into two parts: subsampling on a group *for a specific generator* (Alg. 1), and *how to choose* the generator.

**Algorithm 1** Uniform group subsampling

1: **Input:** Group $G$, Generators $S$, subsampling rate $R$, generator $s_d$
2: **Output:** Subsampled group $G^{\downarrow}$
3: // Get directed Cayley graph
4: $V, E \leftarrow \mathtt{DiCay(G,S)}$
5: $E' \leftarrow E.copy()$
6: **for** each $v \in V$ **do**
7:     // remove generator $s_d$
8:     $E'.remove((v, v \cdot s_d))$
9:     // add generator $s_d^R$
10:     $E'.add((v, v \cdot s_d^R))$
11: **end for**
12: // BFS traversal from $e$
13: $Q \leftarrow \varnothing$
14: $G^{\downarrow} \leftarrow \varnothing$
15: $Q.enqueue(e)$
16: **while** $Q \neq \varnothing$ **do**
17:     $n \leftarrow Q.dequeue()$
18:     $G^{\downarrow}.add(n)$
19:     **for** each $(n, m) \in E'$ **do**
20:         **if** $m \notin Q$ **then**
21:             $Q.enqueue(m)$
22:         **end if**
23:     **end for**
24: **end while**
25: **Return** $G^{\downarrow}$

The key idea behind Alg. 1 is to leverage the structure of the Cayley graph to perform the subsampling. Consider a generating set $S = \{s_1, s_2, \ldots s_n\}$ for a group $G = \langle S \rangle$. Let each generator $s_i \in S$ to have an order $o_i$, *i.e.*, $s_i^{o_i} = e$ and such an order always exists (Isaacs, 2009). We view the uniform subsampling of $G$ by a factor of $R$ for a generator $s_d$ is to uniformly discard elements along the path $(e, s_d^1, s_d^2, \ldots, s_d^{o_d-1})$ on the Cayley graph of $G$. This can also be viewed as adding the generator $s_d^R$ to the generating set $S$ while removing the generator $s_d$. In Example 1, we illustrate the proposed Alg. 1 applied to a sequence.

**Example 1. Discrete-time periodic signal of period** $4$. *The domain corresponds to the translation group on a periodic 1D grid of size 4, with the generator $1$ representing discrete time translation ($\Delta t$). The group action is addition modulo 4, indicating a periodic time shift. Its Cayley graph is shown in Fig. 1 (left). When downsampling by a factor of 2, the generator $\Delta t$ combines to $2\Delta t$. Observe that this is equivalent to the subsampling in Eq. (1) by discarding every other element.*

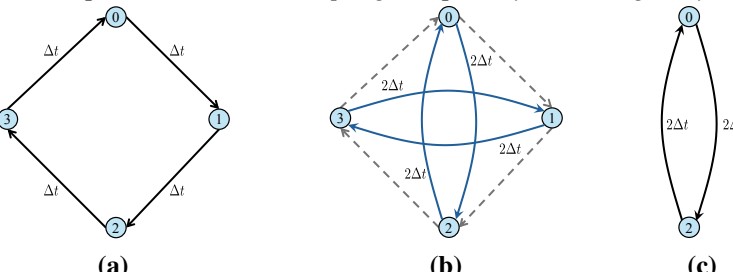

    **(a)**         **(b)**         **(c)**

Figure 1: **(a)** Cayley graph of the group with generator $\Delta t$. **(b)** Edges corresponding to the generator $\Delta t = 1$ are removed (dotted edges) and new edges corresponding to the element $2\Delta t$ are added. **(c)** The resultant cyclic subgroup of size 2 obtain by the traversing the new graph from node 0.

Equipped with the intuition, we now introduce two lemmas before going into the conditions and show why Alg. 1 returns $G^{\downarrow}$ that is a subgroup of $G$.

**Lemma 1.** *For the set $G^\downarrow$ returned by Alg. 1, $v \in G^\downarrow$ if and only if $v$ can be expressed as a product of the elements of the set $S^\downarrow = \left( S/\{s_d\} \right) \cup \{s_d^R\}$.*

**Lemma 2.** *For the set $S^\downarrow$ in Lemma 1, each element $s_i \in S^\downarrow \implies s_i^{-1} \in G^\downarrow$.*

Please see §A3.1 and §A3.2 for the complete proof of the lemmas. With some mild assumptions, using the above lemmas, we can show that the set $G^\downarrow$ returned by Alg. 1 is a *subgroup* of $G$ (see §A3.3). To guarantee that we are indeed downsampling, additional conditions are required such that $G^\downarrow$ is a *proper subset* of $G$, *i.e.*, the size of $G^\downarrow$ is smaller. Specifically, we need conditions to ensure that the discarded group elements cannot be regenerated from the remaining ones.

**Claim 1.** *If $S_d^k = \{s_d^k : k \in \mathbb{Z}^+ \text{ and } k \mod R \not\equiv 0\}$ are non-redundant powers of $s_d$, $o_d \mod r \equiv 0$, and the elements of $S_d^k$ can not be represented as a product of the elements of the left cosets of the subgroup $G_{sub} = \langle S/\{s_d\} \rangle$ generated by the set $\{s_d^{nR} : n \in \mathbb{Z}_0^+\}$ then Alg. 1 returns a proper subgroup $G^\downarrow \subset G$.*

*Proof.* We show that $G^\downarrow$ forms a group by verifying closure (using Lem. 1), the existence of inverses (using Lem. 2), and that associativity and identity hold by construction. We then prove that $G^\downarrow$ is a proper subset of $G$ by showing that, under the assumptions, elements can only be discarded in Alg. 1. The formal proof is provided in §A3.3. □

Clm. 1 imposes conditions that restrict the regeneration of discarded elements by Alg. 1, ensuring a proper subgroup. For a better understanding of the implications of this claim, we provide a visual illustration in §A4. With Alg. 1, we can subsample a group given a specific generator. If there are multiple generators, then different subgroups can be formed. We now discuss how to choose among these subgroups.

**Choice of subgroups.** The choice of subgroup matters. Choosing a generator $s_d$ with a small order to subsample may lead to the complete exclusion of transformations associated with it; see Example 2.

**Example 2. Subsampling of dihedral group** $D_8$. *Here, we illustrate the effect of the choice of generators while subsampling the group $D_8 = \langle s, r | s^2 = r^4 = e, sr = r^3 s \rangle$. While subsampling by a factor of 2, we can subsample along the generator $s$, resulting in a cyclic subgroup of rotation $C_4$. Or, according to our proposed algorithm, we can subsample along $r$, resulting in subgroup $D_4$.*

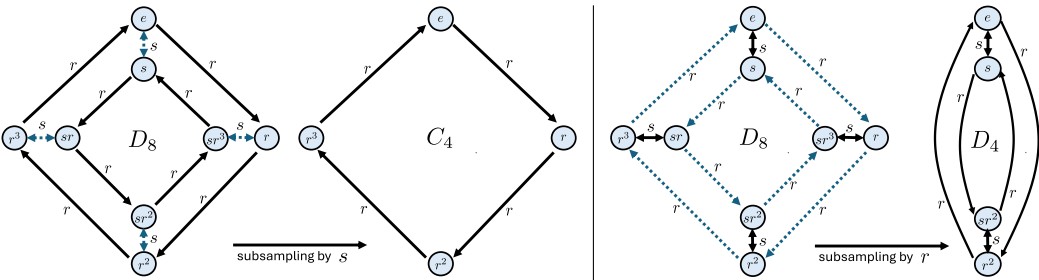

Figure 2: Subsampling group $D_8$ along the generator $s$ (on *left*) and $r$ (on *right*). The edges corresponding to the subsampling generators are dotted in the Cayley graph.

Based on this intuition, we propose a heuristic for selecting a set of generators $D_s$ to subsample $G$ with sampling factors $R_s$ along each generator in $D_s$. Given the subsampling rate $R$, we decompose it into prime factors, *i.e.*, $R = R_1 \cdot R_2 \cdot R_3 \cdots$, sorted in descending order. For each $R_i$, starting from $i = 1$, we select the generator with the maximum order satisfying the constraint outlined in Clm. 1. Subsampling by the factor $R$ can be conceptualized as a sequential subsampling, each by $R_i$. The algorithms for a generalized approach to subsampling and their time complexity analysis are provided in §A5. With subsampling defined, we will next generalize the notion of bandlimited-ness and propose an equivariant anti-aliasing operator to signals on a group.

*Remarks:* The proposed algorithm and heuristic offer a general framework for uniformly subsampling subgroups from any finite group, extending the concept of sampling rate to groups. The heuristic seeks to maximize the number of generators in the subgroup. In practice, choosing the subgroup is a key hyperparameter influenced by the application and may require domain expertise.

## 4.2 THE SUBGROUP SAMPLING THEOREM FOR SIGNALS ON GROUPS

In multi-rate signal processing, the sampling theorem states a sufficient condition (bandlimited-ness) on a signal such that *perfect reconstruction* can be achieved given the signal sampled at a lower rate (Vetterli et al., 2014), *i.e.*, how to sample and interpolate between finite-dimensional vectors. In this section, we propose a sampling theory for signals on finite groups, *i.e.*, a condition that allows for perfect reconstruction from subgroups, and an anti-aliasing filter to ensure that the signal satisfies the condition. We now establish a vectorized notation of the signal to aid the discussion.

Recall, we are considering a signal $x \in \mathcal{X}_G \triangleq \{x : G \to \mathbb{R}\}$, where $G$ is a finite set with size $N$, then $x$ can be equivalently expressed by a finite-dimensional vector $\mathbf{x} \in \mathbb{R}^N$ such that $\mathbf{x}[i] \triangleq x[g_i]$, where $g_i$ denotes the $i^{th}$ element of the group $G$ in an arbitrary fixed order.

Using this notation, the Fourier transform for a finite group $G$ in Eq. (4) can be expressed as a matrix multiplication $\hat{\mathbf{x}} = \mathcal{F}_G \mathbf{x}$, where $\hat{\mathbf{x}} \in \mathbb{C}^N$ denotes the Fourier coefficients. Similarly, the inverse Fourier transform can be expressed as $\mathbf{x} = \mathcal{F}_G^{-1} \hat{\mathbf{x}}$. Note, $\mathcal{F}_G^{-1}$ and $\mathcal{F}_G$ are orthonormal bases.

Next, the sampling operation in Eq. (6) and the interpolation operation can be expressed as matrix multiplications:

$$\text{Sampling: } \mathbf{x}^{\downarrow} = \mathcal{S}\mathbf{x}, \qquad \text{Sampling followed by Interpolation: } \mathbf{x}^{\uparrow} = \mathcal{I}\mathbf{x}^{\downarrow} = \mathcal{I}\mathcal{S}\mathbf{x}, \qquad (7)$$

where $\mathcal{S} \in \mathbb{R}^{M \times N}$ (with $M < N$) is the *sampling matrix* and $\mathcal{I} \in \mathbb{R}^{N \times M}$ denotes the *interpolation matrix*. A *perfect reconstruction* is achieved when $\mathbf{x}^{\uparrow} = \mathbf{x}$, which is not true in general. Eq. (7) describes the standard setup utilized for deriving the Sampling theory for signals on different domains (Vetterli et al., 2014; Chen et al., 2015a)

We now define the sufficient condition, *i.e.*, "*bandlimited*-ness", for signals on groups where perfect reconstruction is possible from signals on the corresponding subgroups.

**Bandlimited functions for subgroup subsampling.** Our main insight is based on the observation that for any bandlimited function $\mathbf{x}$ we need to establish a map $\mathcal{M} \in \mathbb{C}^{N \times M}$ from the Fourier coefficients of the subsampled signal $\hat{\mathbf{x}}^{\downarrow} \triangleq \mathcal{F}_{G^{\downarrow}} \mathbf{x}^{\downarrow}$ to the Fourier coefficients $\hat{\mathbf{x}}$, which results in the following dependencies between

$$\hat{\mathbf{x}} = \mathcal{M}\hat{\mathbf{x}}^{\downarrow} \quad \Rightarrow \quad \mathcal{F}_G^{-1}\hat{\mathbf{x}} = \mathcal{F}_G^{-1}\mathcal{M}\hat{\mathbf{x}}^{\downarrow} \quad \Rightarrow \quad \mathbf{x}^{\downarrow} = \mathcal{S}\mathcal{F}_G^{-1}\mathcal{M}\hat{\mathbf{x}}^{\downarrow}. \qquad (8)$$

Combining Eq. (8) and the fact that $\mathbf{x}^{\downarrow} = \mathcal{F}_{G^{\downarrow}}^{-1}\hat{\mathbf{x}}^{\downarrow}$, we establish the following relationship between $\mathcal{M}$, $\mathcal{S}$ and the Fourier bases:

$$\mathcal{F}_{G^{\downarrow}}^{-1} = \mathcal{S}(\mathcal{F}_G^{-1}\mathcal{M}) = \mathcal{S}\mathcal{B}. \qquad (9)$$

Eq. (9) can be informally viewed as "choosing" a set of vectors $\mathcal{B} \triangleq F_G^{-1}\mathcal{M}$ defined on $G$ such that when subsampled to the subgroup $G^{\downarrow}$, they generate the Fourier basis for the subgroup $G^{\downarrow}$ [1]. Consecutively, we define the interpolation matrix as $\mathcal{I} = \mathcal{B}\mathcal{F}_{G^{\downarrow}}$.

We now state our proposed definition of bandlimited signals in the context of subgroup subsampling.

**Claim 2. Subgroup Sampling Theorem.** *For any signal $\mathbf{x}$ on $G$, if the Fourier coefficients $\hat{\mathbf{x}}$ are in the 1-eigenspace of $\bar{\mathcal{M}} \triangleq \mathcal{M}(\mathcal{M}^{\dagger}\mathcal{M})^{-1}\mathcal{M}^{\dagger}$ then it can be reconstructed perfectly from the subsampled signal $\mathbf{x}^{\downarrow}$ on $G^{\downarrow}$. The superscript $\dagger$ denotes the conjugate transpose.*
*Proof.* To prove the claim, we show that

$$\hat{\mathbf{x}} = \mathcal{M}(\mathcal{M}^{\dagger}\mathcal{M})^{-1}\mathcal{M}^{\dagger}\hat{\mathbf{x}} \quad \Rightarrow \quad \mathbf{x} = \mathcal{B}(\mathcal{B}^{\dagger}\mathcal{B})^{-1}\mathcal{B}^{\dagger}\mathbf{x} \quad \Rightarrow \quad \mathbf{x} = \mathcal{P}_{\mathcal{M}}\mathbf{x}. \qquad (10)$$

Here, $\mathcal{P}_{\mathcal{M}} \triangleq \mathcal{B}(\mathcal{B}^{\dagger}\mathcal{B})^{-1}\mathcal{B}^{\dagger}$ denotes the projection matrix to the column space of $\mathcal{B} \triangleq F_G^{-1}\mathcal{M}$. This means that $\mathbf{x}$ is in $\text{Span}(\mathcal{B})$, *i.e.*, we can express $\mathbf{x} = \mathcal{B}\hat{\mathbf{x}}_c$ for some set of coefficient vector $\hat{\mathbf{x}}_c$. Perfect reconstruction from the subsampled signal $\mathbf{x}^{\downarrow}$ is now possible, *i.e.*,

$$\mathcal{I}\mathbf{x}^{\downarrow} = (\mathcal{B}\mathcal{F}_{G^{\downarrow}})(\mathcal{S}\mathbf{x}) = (\mathcal{B}\mathcal{F}_{G^{\downarrow}}\mathcal{S})(\mathcal{B}\hat{\mathbf{x}}_c) = \mathcal{B}\mathcal{F}_{G^{\downarrow}}\mathcal{F}_{G^{\downarrow}}^{-1}\hat{\mathbf{x}}_c = \mathcal{B}\hat{\mathbf{x}}_c = \mathbf{x}. \qquad (11)$$

The complete proof is provided in §A3.4. $\qquad \qquad \square$

To provide some intuition, let's study how this definition applies to Cyclic groups.

---

[1] The construction of such an $\mathcal{M}$ for an arbitrary group $G$ and its subgroup $G^{\downarrow}$ is nontrivial, as the irreps of the group and the subgroup that constitutes the corresponding Fourier bases often differ in dimensions.

**Example 3. Bandlimited-ness for Cyclic Groups.** *For real-valued functions over the finite cyclic group $C_N$, the real Fourier bases consist of the constant function $\frac{1}{\sqrt{N}}\mathbf{1}$ and $\{\frac{1}{\sqrt{N}}\cos 2\pi k\frac{n}{N}, \frac{1}{\sqrt{N}}\sin 2\pi k\frac{n}{N} : k \leq \lfloor\frac{N-1}{2}\rfloor, \forall n \in \mathbb{Z}/N\mathbb{Z}\}$ where $k$ represents the frequency, and $n \in \mathbb{Z}/N\mathbb{Z}$ represents the elements of $C_N$. If $N$ is even, there is an additional basis $\frac{1}{\sqrt{N}}\cos 2\pi\frac{n}{2}$. Assuming the Fourier coefficients are arranged in an ascending frequency and uniform downsampling by a factor of 2 (with $N \mod 2 = 0$), we have*

$$\mathcal{M} = \sqrt{2}\begin{bmatrix}\mathbf{I}_{\frac{N}{2}}\\ \mathbf{0}_{\frac{N}{2}}\end{bmatrix}, \tag{12}$$

*where $\mathbf{0}_{\frac{N}{2}}$ is a zero matrix of size $\frac{N}{2}\times\frac{N}{2}$, as the Fourier bases of $C_{\frac{N}{2}}$ are formed by sinusoidal of lower-frequencies. The corresponding $\bar{\mathcal{M}} \in \mathbb{C}^{N\times N}$ is:*

$$\bar{\mathcal{M}}_{ij} = \begin{cases}1 & \text{if } i = j \text{ and } i \leq \frac{N}{2}\\ 0 & \text{otherwise}\end{cases}. \tag{13}$$

*The vector $\hat{\mathbf{x}}$ lies in the 1-eigenspace if $\hat{\mathbf{x}}[i] = 0$ for $i > \frac{N}{2}$, aligning precisely with the conventional concept of bandlimited-ness.*

*Remarks:* We have now defined what it means for signals on a finite group to be bandlimited with respect to a given $\mathcal{M}$ that satisfies Eq. (9). To ensure that the signal is bandlimited before subsampling, we can use the projection matrix $\mathcal{P}_\mathcal{M}$ to ensure that the signal satisfies the condition in Clm. 2, *i.e.*, *perform an ideal anti-aliasing*. However, it is easy to observe that the $\mathcal{M}$ is not unique. While many $\mathcal{M}$ achieve perfect reconstruction, they may not be suitable for feature learning. Specifically, the anti-aliasing operation should be equivariant to group actions and preserve some notation of smoothness. We now discuss how to find such an $\mathcal{M}$.

**Equivariant anti-aliasing operator.** We denote the ideal anti-aliasing operator $\mathcal{P}_\mathcal{M}$ in the Fourier space as $\hat{\mathcal{P}}_\mathcal{M} \triangleq \mathcal{F}_G\mathcal{P}_\mathcal{M}\mathcal{F}_G^{-1}$. Our goal is to find a $\mathcal{M}$ that achieves perfect reconstruction, performs an equivariant anti-aliasing operation, and extracts smooth features. We formulate this goal as an optimization problem:

$$\mathcal{M}^* = \underset{\mathcal{M}}{\arg\min}\underbrace{\left\|\text{vec}(\hat{\mathcal{P}}_\mathcal{M}) - \bar{\mathbf{T}}\text{vec}(\hat{\mathcal{P}}_\mathcal{M})\right\|_2^2}_{\text{\textcolor{teal}{Equivariance Objective}}} + \lambda\underbrace{\mathbf{1}^\top\left(\text{Diag}(\mathcal{F}_G^{-1\dagger}\mathbf{L}\mathcal{F}_G^{-1})\mathcal{M}^{|\cdot|}\right)^\top}_{\text{\textcolor{teal}{Smooth Selection Objective}}} \tag{14}$$

$$\text{subject to } \mathcal{F}_{G\downarrow}^{-1} = \mathcal{S}\mathcal{F}_G^{-1}\mathcal{M} \text{ (\textcolor{orange}{Perfect Reconstruction Constraint})}.$$

Here, $\lambda > 0$ is a hyperparameter balancing equivariance and smoothness, the superscript $|\cdot|$ denotes the elementwise absolute value, $\text{Diag}$ returns the diagonal elements as a row vector and the details of $\bar{\mathbf{T}}$ and $\mathbf{L}$ are described below.

To be an $G$-equivariant the anti-aliasing operator, $\mathcal{P}_\mathcal{M}$ needs to satisfy the following equivariant constraint:

$$\hat{\mathcal{P}}_\mathcal{M}\hat{\rho}_{\mathcal{X}_G}(g)\hat{\mathbf{x}} = \hat{\rho}_{\mathcal{X}_G}(g)\hat{\mathcal{P}}_\mathcal{M}\hat{\mathbf{x}}, \quad \forall g \in G, \hat{\mathbf{x}} \in \mathbb{C}^n. \tag{15}$$

Here, we describe the equivariance constraint in the Fourier domain where $\hat{\rho}_{\mathcal{X}_G}(g)$ corresponds to the action of the group $G$ on the Fourier coefficients formed by the direct sum of the corresponding *irreps* (see §A1.2 for details).

Next, Mouli & Ribeiro (2021) show that linear operators that are contained within the 1-eigenspace of the Reynolds operator $\bar{\mathbf{T}}$ corresponding to the tensor product representation

$$\hat{\rho}_{\mathcal{X}_G\otimes\mathcal{X}_G} = \hat{\rho}_{\mathcal{X}_G}(g) \otimes \hat{\rho}_{\mathcal{X}_G}(g^{-1})^\top \tag{16}$$

satisfy Eq. (15), *i.e.*, are equivariant, where

$$\bar{\mathbf{T}} \triangleq \frac{1}{G}\sum_{g\in G}\hat{\rho}_{\mathcal{X}_G}(g) \otimes \hat{\rho}_{\mathcal{X}_G}(g^{-1})^\top. \tag{17}$$

Hence, the equivariance constraint of $\hat{\mathcal{P}}_\mathcal{M}$ can be written as $\text{vec}(\hat{\mathcal{P}}_\mathcal{M}) = \bar{\mathbf{T}}\text{vec}(\hat{\mathcal{P}}_\mathcal{M})$ (see §A1.3). Finally, we relax this equality condition as a penalty term to form the *equivariance objective* in Eq. (14).

Next, the *smooth selection objective* is designed to prefer smoother basis functions in constructing the bandlimited subspace. To quantify the smoothness of signals over groups, we view them as functions over their corresponding Cayley graphs. We adopt the notion of smoothness from graph signal processing, namely, the *Laplacian quadratic form* (Dong et al., 2016; Shuman et al., 2013) as the smoothness measure. The Laplacian quadratic form for a function $f$ on $G$ can be defined as $f^\top \mathbf{L} f$, where $\mathbf{L}$ is the Laplacian of the Cayley graph $\Gamma(G, S)$. A smaller value indicates a smoother function. Intuitively, the smooth selection objective can be viewed as penalizing the Fourier bases by their Laplacian quadratic form weighted by their corresponding elements in $\mathcal{M}^{|\cdot|}$.

Finally, we solve the constrained optimization problem in Eq. (14) via Sequential Least Squares Programming (Kraft, 1988) to obtain $\mathcal{M}^\star$ which defines the bandlimited-ness and a corresponding anti-aliasing operator $\mathcal{P}_{\mathcal{M}^\star}$.

**Anti-aliased $G$-CNN.** In group equivariant CNN (Cohen & Welling, 2016), the input is first transformed to functions/features over the desired group. When performing subgroup subsampling, our designed subsampling and anti-aliasing operator are applied to these functions in the group. We discuss this operation in detail in Appendix §A6.

## 5 EXPERIMENTS AND EVALUATIONS

### 5.1 EMPERICAL VALIDATION FOR CLAIM 2

We validate our theoretical findings in Clm. 2 by numerically checking the recovery of bandlimited functions after subsampling. We generate random signals $\mathbf{x}$ defined on dihedral group $D_{2n} = \langle s, r | s^2 = r^n = (sr)^2 = e \rangle$ and cyclic rotation group $C_n = \langle r | r^n = e \rangle$, sampling each value from the standard Gaussian $\mathcal{N}(0, 1)$. We consider subgroups $G^\downarrow$, then apply the proposed downsampling technique: project $\mathbf{x}$ onto a bandlimited subspace by $\tilde{\mathbf{x}} = \mathcal{P}_{\mathcal{M}} \mathbf{x}$ (anti-aliasing) and obtain $\mathbf{x}^\downarrow$ restricted to $G^\downarrow$ using $\mathcal{S}$ (subsampling). Lastly, we interpolate the downsampled signal to the original group using $\mathbf{x}^\uparrow = \mathcal{I} \mathbf{x}^\downarrow$.

Table 1: Empirical Validation of Claim 2. We report the recon. error with / (and without) the anti-aliasing operation. Anti-aliasing achieves zero recon. error up to numerical precision.

| Group | Subgroup | Sub. R. | Recon. Err. |
|---|---|---|---|
| | $D_{14}$ | 2 | 1.72e-13/3.8 |
| $D_{28}$ | $C_{14}$ | 2 | 6.54e-13/4.0 |
| | $C_7$ | 4 | 9.48e-14/5.2 |
| | $D_{10}$ | 2 | 4.10e-11/3.3 |
| $D_{20}$ | $C_{10}$ | 2 | 3.03e-11/3.4 |
| | $D_4$ | 5 | 2.78e-14/4.7 |
| $C_{30}$ | $C_{15}$ | 2 | 5.18e-13/4.2 |
| | $C_5$ | 6 | 9.54e-14/5.9 |

In Tab. 1, we report reconstruction error, defined the norm difference $\|\tilde{\mathbf{x}} - \mathbf{x}^\uparrow\|_2^2$ between the bandlimited signal ($\tilde{\mathbf{x}}$ and the interpolated signal $\mathbf{x}^\uparrow$). We observe that the interpolation operator successfully reconstructs the bandlimited signal. To further study the proposed anti-aliasing operator, we visualize its response to the unit sample function $\delta_G[g]$, where $\delta_G[g] = 1$ if $g = e$ and 0 otherwise. This response to $\delta_G$ represents the smoothing filter used in anti-aliasing. In Fig. 3, we illustrate such filters. We observe that for the downsampling of the cyclic group ($C_{16}$ to $C_8$), the filter is reminiscent of the sinc function (Fig. 3), which is used in an ideal low-pass filter for sequences. This further illustrates the relation of our anti-aliasing to the classic anti-aliasing on sequences as explained in Example 3.

*Remarks:* In practice, ideal anti-aliasing operators are often approximated. For instance, the Gaussian blur filter is commonly used to smooth signals, approximating the sinc function, which has better empirical advantages. Building on our theorem, there is potential for developing a more efficient smoothing filter directly in the group ("time") domain.

### 5.2 IMAGE CLASSIFICATION

We apply the proposed subgroup selection and anti-aliasing operator to equivariant CNN architectures. Note that, to use the proposed anti-aliasing filter $\mathcal{P}_{\mathcal{M}}$ in deep nets, we only need to perform the optimization in Eq. (14) *only once* before training a model.

**Experiment setup.** As in prior works (Cesa et al., 2021; Cohen & Welling, 2017), we study the effects of subgroup subsampling and anti-aliasing on group equivariant classification models using the rotated MNIST (Deng, 2012) and CIFAR10 (Krizhevsky et al., 2009).

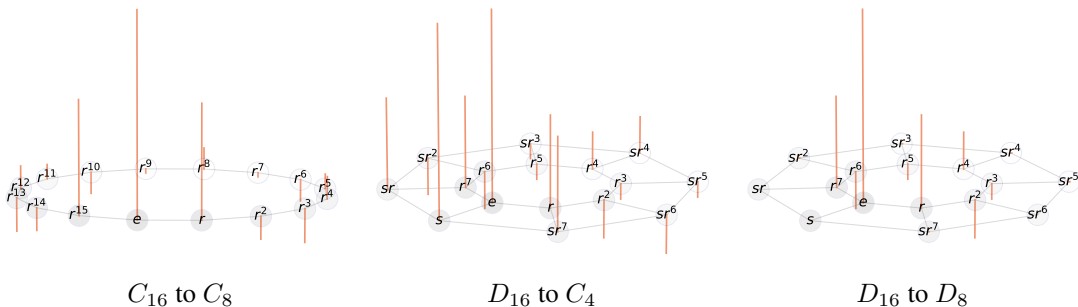

$$C_{16} \text{ to } C_8 \qquad\qquad D_{16} \text{ to } C_4 \qquad\qquad D_{16} \text{ to } D_8$$

Figure 3: Visualization of the smoothing filter $(\mathcal{P}_{\mathcal{M}}\delta_G)$ used in the anti-aliasing operation for subgroup subsampling. The vertical bar corresponds to the value of the filter at each node, with the downward bars indicating negative values.

Table 2: Performance of $G$-equivariant models on Rotated MNIST and CIFAR-10 at different subsampling rates $R$ and with/without anti-aliasing filter $\mathcal{P}_{\mathcal{M}^\star}$ under the continuous rotation and roto-reflection symmetry $(SO(2)/O(2))$. Sub-group subsampling with anti-aliasing improves both equivariance and accuracy.

| | $R$ | # Param. $\times 10^3$ | $\mathcal{P}_{\mathcal{M}^\star}$ | Sym. $(SO(2))$ | | | | Sym. $(O(2))$ | | | |
|---|---|---|---|---|---|---|---|---|---|---|---|
| | | | | $\text{Acc}_{\text{no aug}}$ | $\text{Acc}_{\text{loc}}$ | $\text{Acc}_{\text{orbit}}$ | $\mathcal{L}_{\text{equi}}$ | $\text{Acc}_{\text{no aug}}$ | $\text{Acc}_{\text{loc}}$ | $\text{Acc}_{\text{orbit}}$ | $\mathcal{L}_{\text{equi}}$ |
| MNIST | - | 323.11 | - | 0.9767 | 0.8234 | 0.8346 | 0.058 | 0.9752 | 0.8253 | 0.8496 | 0.039 |
| | 2 | 194.09 | ✗ | 0.9743 | 0.8007 | 0.8106 | 0.056 | 0.9774 | 0.6878 | 0.5660 | 0.092 |
| | 2 | 194.09 | ✓ | 0.9773 | 0.8301 | 0.8358 | 0.049 | 0.9807 | 0.6976 | 0.5749 | 0.091 |
| | 3 | 151.08 | ✗ | 0.9674 | 0.7762 | 0.7907 | 0.057 | 0.9731 | 0.8044 | 0.8316 | 0.046 |
| | 3 | 151.08 | ✓ | 0.9731 | 0.8057 | 0.8173 | 0.047 | 0.9724 | 0.8251 | 0.8451 | 0.037 |
| | 4 | 129.57 | ✗ | 0.9831 | 0.6283 | 0.5052 | 0.109 | 0.9810 | 0.6614 | 0.4816 | 0.109 |
| | 4 | 129.57 | ✓ | 0.9827 | 0.6547 | 0.5219 | 0.093 | 0.9806 | 0.6978 | 0.5006 | 0.098 |
| CIFAR-10 | - | 549.33 | - | 0.6934 | 0.4253 | 0.3708 | 0.322 | 0.7251 | 0.4463 | 0.3867 | 0.265 |
| | 2 | 291.29 | ✗ | 0.7060 | 0.4659 | 0.4096 | 0.398 | 0.7448 | 0.4757 | 0.3310 | 0.555 |
| | 2 | 291.29 | ✓ | 0.7088 | 0.4868 | 0.4279 | 0.336 | 0.7418 | 0.4720 | 0.3274 | 0.460 |
| | 3 | 205.27 | ✗ | 0.7006 | 0.4337 | 0.3766 | 0.549 | 0.7249 | 0.4210 | 0.3674 | 0.478 |
| | 3 | 205.27 | ✓ | 0.6945 | 0.4472 | 0.3876 | 0.379 | 0.7117 | 0.4794 | 0.4197 | 0.411 |
| | 4 | 162.26 | ✗ | 0.7075 | 0.4275 | 0.2866 | 0.625 | 0.7590 | 0.5205 | 0.2921 | 0.607 |
| | 4 | 162.26 | ✓ | 0.7000 | 0.4536 | 0.3091 | 0.439 | 0.7525 | 0.5425 | 0.3017 | 0.550 |

To rigorously examine the impact of subsampling on the rotation and roto-reflection symmetry preservation, we remove the digits '9', '2', and '4' from MNIST following Wang et al. (2023). These digits disrupt the symmetry assumption on the dataset that the labels remain unchanged under the group action. For instance, the digit '6' overlaps with '9' under a $180°$ rotation. The same is true for '2'/ '5' and '4'/'7' under the roto-reflection group.

While we consider the symmetry of the image data to be continuous rotation/roto-reflection $(SO(2)/O(2))$, note that we use the discrete $C_{24}$ and $D_{24}$ equivariant CNN for computational feasibility which matches our theoretical assumptions. For MNIST and CIFAR-10, we train on 5k and 60k training images, and test on images on different levels of transformations (see §A7 for details).

**Evaluation metrics.** We propose evaluation metrics to measure the equivariance and classification performance. Given a deep-net $H$, we use the features of image $x$ to measure *equivariance error*

$$\mathcal{L}_{\text{Equi.}}(x) = \frac{1}{|G|} \sum_{g \in G} \frac{\|H(\rho_{\text{In}}(g)x) - \rho_{\text{Out}}(g)H(x)\|_2^2}{\|H(x)\|_2^2}. \tag{18}$$

The representations $\rho_{\text{In}}$ and $\rho_{\text{Out}}$ correspond to the group action on the input and output space of the deep-net $H$. In the context of image classification, $\rho_{\text{IN}}$ represents the action of 2D rotation (and flipping), while $\rho_{\text{OUT}}$ remains as an identity, *i.e.*, invariance. Specifically, we use the pooled features from the final equivariant convolution layer as the invariant features (Weiler et al., 2018).

For the classification performance, we consider *three accuracy metrics* for evaluating the model performance under different degrees of equivariance:

Table 3: Impact of subgroup selection in subgroup sampling on a 3-layer equivariant CNN. "*" indicates selection based on our method. Our algorithm improves performance for various sampling rates.

| Group | Sub. R. | Subgroups | $\texttt{Acc}_{\texttt{no aug}}$ | $\texttt{Acc}_{\texttt{loc}}$ | $\texttt{Acc}_{\texttt{orbit}}$ |
|---|---|---|---|---|---|
| $D_{24}$ | $1, 2, 2$ | $D_{24}\ C_{12}\ C_6$ | 0.9703 | 0.6215 | **0.6128** |
| | | $D_{24}\ D_{12}\ D_6$* | **0.9726** | **0.6539** | 0.5489 |
| $D_{24}$ | $1, 4, 1$ | $D_{24}\ C_6\ C_6$ | 0.9766 | 0.5244 | 0.4596 |
| | | $D_{24}\ D_6\ D_6$* | **0.9767** | **0.6272** | **0.4860** |
| $D_{28}$ | $1, 2, 1$ | $D_{28}\ C_{14}\ C_{14}$ | 0.9742 | 0.5852 | 0.5191 |
| | | $D_{28}\ D_{14}\ D_{14}$* | **0.9786** | **0.7085** | **0.5792** |

1. $\texttt{Acc}_{\texttt{no aug}}$: The accuracy of the model on the original (un-augmented) dataset.

2. $\texttt{Acc}_{\texttt{loc}}$: The accuracy of the model on the "locally augmented" dataset.

3. $\texttt{Acc}_{\texttt{orbit}}$: The accuracy of the model on the full $(SO(2)/O(2))$ orbit of the dataset.

The orbit is constructed by taking all $10°$ rotations, and for local augmentations, we report on random rotations within $\pm 60°$ of the test set.

**Results.** In Tab. 2, we report the results for MNIST and CIFAR-10 datasets under $SO(2)$ and $O(2)$ symmetry. We report the average over 3 runs. For all models, the standard deviations are: $\texttt{Acc}_{\texttt{no aug}} < 0.001$, $\texttt{Acc}_{\texttt{orbit}}$ and $\texttt{Acc}_{\texttt{loc}} < 0.01$, and $\mathcal{L}_{\texttt{equi}} < 0.004$.

Overall, we observe that subgroup subsampling significantly reduces the parameter count of the equivariant models. However, increasing the sampling rate (*e.g.*, $R = 4$) disrupts the strict equivariance constraint, leading to a higher equivalence error ($\mathcal{L}_{\texttt{equi}}$). This manifests as a decrease in both $\texttt{Acc}_{\texttt{orbit}}$ and $\texttt{Acc}_{\texttt{loc}}$, while increasing the accuracy on the original test set, $\texttt{Acc}_{\texttt{no aug}}$. Next, incorporating our anti-aliasing operation mitigates the invariance error and achieves higher $\texttt{Acc}_{\texttt{orbit}}$ and $\texttt{Acc}_{\texttt{loc}}$. Notably, a lower sampling rate combined with appropriate anti-aliasing *significantly reduces parameter* usage while maintaining comparable or even surpassing the accuracy and equivariance achieved with the full equivariant models.

We provide additional results of our model on STL-10 (Coates et al., 2011) dataset, where we also observe similar performance gain (see Appendix §A2.2).

**Ablations.** In Tab. 3, we provide the ablation of the proposed sub-group selection heuristic. For 3 layered equivariant CNN and different sampling rates at different layers of the models, we report the accuracy metrics for different choices of subgroups. We observe that for different symmetry groups at different sampling rates, our proposed subgroup selection improves the performance in most cases.

Furthermore, in §A2.1, we demonstrate index selection by Xu et al. (2021) can be used with our technique. The results show further performance improvements and confirm that our method can easily be incorporated with existing techniques.

**Limitations.** As this is a theoretical paper, we fully acknowledge that our experiments are limited to small-scale datasets and models. These experiments are meant to study and demonstrate the potential of the proposed framework. Our proposed downsampling layer currently operates on finite groups rather than continuous ones. The time complexity of the subgroup selection algorithm scales quadratically, in the worst case, with the number of edges, $|E|$, in the Cayley graph (see §A5).

## 6    CONCLUSION

We propose uniform subgroup downsampling for signals on finite groups with an equivariant anti-aliasing operation. We generalize the uniform subsampling operation to groups and propose a subgroup selection method based on maximizing the number of generators. We then extend the sampling theorem to subgroup subsampling, generalizing the notion of bandlimited-ness and anti-aliasing to groups. We apply these theories to equivariant CNN and empirically show that models with subgroup subsampling can achieve comparable or even better performance compared to full equivariant models. In summary, we believe our developed theory would serve as the foundation for future research in equivariant deep nets and signal processing on groups. We are particularly excited about how to find an "optimal" subgroup for a given task and how to design more effective anti-aliasing for signals on groups that would build on top of our framework.

ACKNOWLEDGMENT

The authors would like to thank Renan A. Rojas-Gomez for providing feedback on the draft version of this work, which helped to improve clarity.

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

APPENDIX

The appendix is organized as follows:

- In §A1, we present a review on group theory.

- In §A2, we present the results for incorporating the equivariant index selection operation with our proposed downsampling technique.

- In §A3, we provide the complete proofs of the Lemmas and Claims in the main paper.

- In §A4, we provide the illustration of the implications of Claim 1.

- In §A5, we provide a further generalization of the approach and how to check whether a given group satisfies our theoretical assumptions.

- In §A7, we document additional implementation details. Code is also provided in the supplemental materials.

## A1 GROUP THEORY PRELIMINARIES

### A1.1 GROUP

A Group is a set $G$ equipped with an operation $\langle \cdot \rangle$ that maintains the following properties:

- **Closure:** $\forall a, b \in G, a \cdot b \in G$

- **Associativity:** $\forall a, b, c \in G, a \cdot (b \cdot c) = (a \cdot b) \cdot c$

- **Existence of Identity:** $\exists e \in G : \forall a \in G, \ e \cdot a = a$

- **Existence of Inverse:** $\forall a \in G, \exists a^{-1} \in G : a \cdot a^{-1} = e$.

$H$ is a subgroup of $G$ if $H \subseteq G$ and $H$ satisfies all the group properties. The cardinality of the set $G$ is known as the ***order of the group***. And the groups of finite order are called ***finite groups***. For any subgroup $H \subseteq G$, the ***left coset*** generated an element $g \in G$ is denoted as $gH = \{gh : h \in H\}$ and ***right coset*** is denoted as $Hg = \{hg : h \in H\}$.

**Discrete Rotation Group** ($C_n$). The discrete rotation group $C_n = \langle r \mid r^n = e \rangle$ is a cyclic group representing rotations by integer multiples of $\frac{360°}{n}$ degrees.

**Dihedral Group.** Dihedral group $D_{2n} = \langle s, r | s^2 = r^n = (sr)^2 = e \rangle$ is the group of symmetries of a regular $n$-sided polygon, with rotations by integer multiple of $\frac{360°}{n}$ and horizontal reflection.

**General Linear Group.** The *general linear group* $\mathrm{GL}(n, \mathbb{F})$ is the group of all invertible $n \times n$ matrices with entries from the field $\mathbb{F}$. $\mathrm{GL}(V)$ denotes general linear group on the vector space $V$.

**Minimal Generation Set.** In general a group $G$ can have multiple minimal generating sets. For example, for any cyclic group $C_N$ generated by the minimal generating set $S = \{r\}$, $S' = \{r^b\}$ is also a minimal generating set when $b$ and $N$ are relatively prime.

### A1.2 GROUP REPRESENTATION

**Linear Group Representation.** A *linear group representation* of a group $G$ on a vector space $U$ is a homomorphism $\rho$ from $G$ to the general linear group $\mathrm{GL}(U)$. This can be written as:

$$\rho : G \to \mathrm{GL}(U), \tag{A19}$$

where for each $g \in G$, $\rho(g)$ is an invertible linear transformation on $U$.

The map $\rho$ must satisfy the following properties :

- $\rho(gh) = \rho(g)\rho(h)$ for all $g, h \in G$.

- $\rho(e) = \mathbf{I}_U$, where $\mathbf{I}_U$ denotes identity transformation on $U$.

- $\rho(g)$ is an invertible linear transformation.

The dimensionality of a representation $\rho$ is equal to the dimensionality of $U$ and written as $d_\rho$.

The **trivial representation** of a group $G$ on a vector space $U$ is a representation $\rho$ such that $\rho(g) = \mathbf{I}_U$ for all $g \in G$. In other words, every group element acts as the identity transformation on the vector space $U$.

The **regular representation** of a finite group $G$ on the vector space $\mathbb{F}^{|G|}$ with a basis indexed by elements of $G$ act on it by permuting these basis elements according to the group operation.

**Equivalent Representation** Two representations $\rho_1$ and $\rho_2$ are equivalent iff $\rho_1 = \mathbf{T}\rho_2\mathbf{T}^{-1}$ for some change of basis $\mathbf{T} \in \mathrm{GL}(U)$.

**Direct Sum of Representations** Let $\rho_1 : G \to \mathrm{GL}(U_1)$ and $\rho_2 : G \to \mathrm{GL}(U_2)$ be two representations of a group $G$ on vector spaces $U_1$ and $U_2$ over the field $\mathbb{F}$. The *direct sum* of $\rho_1$ and $\rho_2$ is a representation $\rho_1 \oplus \rho_2 : G \to \mathrm{GL}(U_1 \oplus U_2)$ defined by:

$$(\rho_1 \oplus \rho_2)(g) = \rho_1(g) \oplus \rho_2(g) \quad \text{for all } g \in G,$$

where $U_1 \oplus U_2$ is the direct sum of $U_1$ and $U_2$, and the action on $U_1 \oplus U_2$ is given by:

$$[\rho_1 \oplus \rho_2](g)[u_1, u_2] = [\rho_1(g)u_1, \rho_2(g)u_2]$$

for all $u_1 \in U_1$ and $u_2 \in U_2$.

**Irreducible Representation.** A representation $\rho : G \to \mathrm{GL}(U)$ of a group $G$ on a vector space $U$ over a field $\mathbb{F}$ is called *irreducible* if the only $G$-invariant subspaces of $U$ are the trivial subspace $\{\mathbf{0}\}$ and $U$.

The set of irreducible representations (irreps) of $G$ is denoted as $\hat{G}$. All the irreps of abelian groups are 1 dimensional. The irreps of dihedral groups are 1 and 2 dimensional. And the irreps of a symmetric group $S_4$ are 1,2,3 dimensional.

**Orthogonality Relation and Fourier Transform.** Let $\varphi$ be an irreducible unitary representation of $G$ of degree $d_\varphi$. Then the $d_\varphi^2$ functions $\{\sqrt{d_\varphi}\varphi^{mn} \mid 1 \leq m, n \leq d_\varphi\}$ form an orthonormal set.

Let $G$ be a finite group. Let $\hat{G} = \{\varphi_i, \ldots, \varphi_s\}$ be a complete set of irreducible representations of $G$. Then the functions

$$\left\{ \sqrt{d_{\varphi_i}}\varphi_i^{mn} \mid \varphi_i \in \hat{G}, 1 \leq m, n \leq d_{\varphi_i} \right\}$$

form an orthonormal set in the space of complex-valued functions over group $L(G)$, and $d_{\varphi_1}^2 + \cdots + d_{\varphi_s}^2 = |G|$. In fact, this set of orthonormal bases defined the Fourier basis for functions on the group $G$. The Fourier transform of a square-integrable function $f \in L^2(G)$ is

$$\hat{f}(\varphi_i^{mn}) = \frac{1}{|G|} \sum_{g \in G} \sqrt{d_{\varphi_i}} f(g) \overline{\varphi_i^{mn}(g)} \ \ \forall \varphi_i \in \hat{G} \text{ and } 1 \leq m, n \leq d_{\varphi_i} \tag{A20}$$

where $\varphi_i^{mn}(g)$ denotes the entry at $m^{\text{th}}$ row and $n^{\text{th}}$ column for matrix $\varphi_i(g)$. Next, $\hat{f}(\varphi_i^{mn})$ denotes the **Fourier coefficient** corresponding to irrep component $\varphi_i^{mn}$. Similarly, the inverse Fourier transform on a group can be expressed as

$$f(g) = \sum_{\varphi_i \in \hat{G}} \sum_{mn \leq d_{\varphi_i}} \hat{f}(\varphi_i^{mn}) \sqrt{d_{\varphi_i}} \varphi_i^{mn}(g), \tag{A21}$$

And if for any $g \in G$ the action on $f \in L^2(G)$ is defined as $[g \cdot f](u) = f(g^{-1}u)$ then the action can be represented in Fourier space as

$$\widehat{g \cdot f}(\varphi_i) = \varphi_i \hat{f}(\varphi_i) \tag{A22}$$

where, $\varphi_i, \ \hat{f}(\varphi_i), \ \widehat{g \cdot f}(\varphi_i) \ \in \mathbb{C}^{d_{\varphi_1} \times d_{\varphi_i}}$.

In the real case, irreps can have redundant columns. To eliminate redundancy, an *endomorphism basis* $C_{\psi_i}$ is constructed to span the non-redundant columns of an irrep $\psi_i$. The full irrep can be recovered by multiplying these columns with elements of $C_{\psi_i}$. The reverse of this process can also be constructed, giving us the non-redundant columns (see Cesa et al. (2021) for details).

### A1.3 INVARIANT AND EQUIVARIANT MAPS

A linear map $\mathcal{W} \in \mathbb{R}^{n' \times n}$ is equivariant with respect to the group action $\rho_U : g \to GL(U)$ and $\rho_{U'} : g \to GL(U')$ with $U \subseteq \mathbb{R}^n$ and $U' \subseteq \mathbb{R}^{n'}$ if

$$\mathcal{W}\rho_U(g)v = \rho_{U'}(g)\mathcal{W}v \ \ \forall g \in G, \forall v \in U \tag{A23}$$

This imposes the following restrictions on the linear map

$$\rho_{U'}(g) \otimes \rho_U(g^{-1})^\top \text{vec}(\mathcal{W}) = \text{vec}(\mathcal{W}) \ \ \forall g \in G \tag{A24}$$

where vec denotes vectorization operation converting a matrix to a vector. The condition in Eq. (A24) denotes that $\text{vec}(\mathcal{W})$ should be invariant to action of tensor product representation $\rho_{U' \otimes U} = \rho_{U'}(g) \otimes \rho_U(g^{-1})^\top$ on the left. For a finite group $G$, the Reynolds operator (Mouli & Ribeiro, 2021; Mumford et al., 1994) is defined as

$$\mathbf{T} = \frac{1}{|G|} \sum_{g \in G} \rho(g) \tag{A25}$$

which is a G-invariant linear map with respect to the representation $\rho : g \to GL(\mathcal{X})$ on vector space $\mathcal{X}$. And to satisfy the condition in Eq. (A24), $\text{vec}(\mathcal{W})$ must belong to the 1-eigenspace of Reynolds operator with respect to the tensor product representation $\rho_{U' \otimes U}$ acting on the vector space $U' \otimes U$ (Mouli & Ribeiro, 2021).

### A1.4 ILLUSTRATION CHALLENGES IN SUBGROUP SUBSAMPLING

To illustrate the challenges in subgroup subsampling, we present an example by subsampling the group

$$D_6 = \{e, r, r^2, s, sr, sr^2\}, \tag{A26}$$

which is a relatively small group of size 6 by *a factor of* 2, *i.e.*, we aim to discard every other element.

As can be seen, discarding every other element of the set will generate the subset $H = \{e, r^2, sr\}$. We can see that $H$ is not a subgroup as the inverse of $r^2$ is missing in the subset; thus, it violets the property of a group.

Additionally, in the above example, we first assumed an ordering of elements of $D_6$. However, we can also choose another ordering of the elements as

$$D_6 = \{r, e, r^2, s, sr, sr^2\}, \tag{A27}$$

resulting in a different subset after subsampling. In this example, even with a small group, the set can be arranged in 720 in different ways, creating ambiguity in the process. So, the traditional subsampling operation of naively discarding elements does not apply to groups.

## A2 ADDITIONAL EXPERIMENTS

### A2.1 EQUIVARIANT SUBSAMPLING

The work Xu et al. (2021) proposes to select indexes in a consistent manner that respects a specialized equivariance (see Xu et al. (2021) lemma 2.1) that can work between groups and subgroups, which is equivalent to Chaman & Dokmanic (2021) in traditional subsampling. The work assumes that the subgroup is already provided and does not perform any anti-aliasing. However, the equivariant index selection scheme can be incorporated with our proposed subgroup selection and anti-aliasing operator. In Tab. A1, we provide the results on rotated-MNIST ($SO(2)$ symmetry), where we incorporate equivariant index selection with our proposed subgroup selection and anti-aliasing operator. We observe that the proposed anti-aliasing operator consistently reduces the equivariance error and improves accuracy. It also demonstrates the wide applicability of our proposed method.

Table A1: Performance of G-equivariant models on Rotated MNIST at different subsampling rates and with/without anti-aliasing filter $\mathcal{P}_{\mathcal{M}^\star}$ with **the equivariant index selection**. We can observe that our proposed technique improves performance and equivariance error, showing wide adaptability.

| Sub. R. | # Param.$\times 10^3$ | $\mathcal{P}_{\mathcal{M}^\star}$ | $\mathtt{Acc_{no\ aug}}$ | $\mathtt{Acc_{loc}}$ | $\mathtt{Acc_{orbit}}$ | $\mathcal{L}_{\mathtt{equi}}$ |
|---|---|---|---|---|---|---|
| 2 | 194.09 | ✗ | 0.9733 | 0.8147 | 0.8225 | 0.0545 |
| 2 | 194.09 | ✓ | 0.9782 | 0.8288 | 0.8297 | 0.0473 |
| 3 | 151.08 | ✗ | 0.9692 | 0.7717 | 0.7865 | 0.1061 |
| 3 | 151.08 | ✓ | 0.9650 | 0.7737 | 0.7833 | 0.0594 |
| 4 | 129.57 | ✗ | 0.9656 | 0.6606 | 0.5602 | 0.0881 |
| 4 | 129.57 | ✓ | 0.9703 | 0.6928 | 0.5759 | 0.0761 |

## A2.2 RESUTL ON STL-10

We conduct experiment on STL-10 (Coates et al., 2011) dataset. This dataset comprises images with a resolution of $96 \times 96$ from 10 different object classes. We train on 5000 training images with any data augmentation. In Tab. A2 and Tab. A3, we present the result on rotation and roto-reflection symmetry groups, respectively, at different sampling rates. We observe that our proposed method achieves higher accuracy and lower equivariance error compared to naive subsampling. We also observe that, since the images in STL-10 have a much higher resolution compared to MNIST and CIFAR-10 and contain more high-frequency details, our anti-aliasing operator provides a higher performance improvement.

Table A2: Performnace of G-equivarinat models on STL-10 dataset at different sampling rate $R$ and with/without anti-aliasing filter $\mathcal{P}_{\mathcal{M}^\star}$ under the rotation ($SO(2)$) symmetry. Sub-group subsampling with anti-aliasing improves both equivariance and accuracy.

| Initial Group | Sub. R. | $\mathcal{P}_{\mathcal{M}^*}$ | #params | $\mathtt{ACC_{no\ aug}}$ | $\mathtt{ACC_{loc}}$ | $\mathtt{ACC_{orbit}}$ | $\mathcal{L}_{\mathtt{equi}}$ |
|---|---|---|---|---|---|---|---|
| $C_{24}$ | - | - | 1.3M | 0.54 | 0.34 | 0.30 | 0.16 |
| $C_{24}$ | 2 | ✓ | 962K | 0.60 | 0.42 | 0.37 | 0.16 |
| $C_{24}$ | 2 | ✗ | 962K | 0.60 | 0.40 | 0.35 | 0.17 |
| $C_{24}$ | 3 | ✓ | 831K | 0.62 | 0.42 | 0.37 | 0.16 |
| $C_{24}$ | 3 | ✗ | 831K | 0.60 | 0.38 | 0.34 | 0.18 |

Table A3: Performance of $G$-equivariant models on STL-10 dataset at different sampling rates $R$ and with/without anti-aliasing filter $\mathcal{P}_{\mathcal{M}^\star}$ under the roto-reflection ($O(2)$) symmetry. Sub-group subsampling with anti-aliasing improves both equivariance and accuracy.

| Initial Group | Sub. R. | $\mathcal{P}_{\mathcal{M}^*}$ | #params | $\mathtt{ACC_{no\ aug}}$ | $\mathtt{ACC_{loc}}$ | $\mathtt{ACC_{orbit}}$ | $\mathcal{L}_{\mathtt{equi}}$ |
|---|---|---|---|---|---|---|---|
| $D_{24}$ | - | - | 1.3M | 0.57 | 0.37 | 0.32 | 0.12 |
| $D_{24}$ | 2 | ✓ | 962K | 0.64 | 0.40 | 0.27 | 0.19 |
| $D_{24}$ | 2 | ✗ | 962K | 0.61 | 0.40 | 0.26 | 0.20 |
| $D_{24}$ | 3 | ✓ | 831K | 0.64 | 0.44 | 0.39 | 0.17 |
| $D_{24}$ | 3 | ✗ | 831K | 0.60 | 0.33 | 0.33 | 0.17 |

## A2.3 EQUIVARIANCE ERROR PROPAGATION

To further study the effect of the anti-aliasing operator, we visualize the propagation of the equivariance error through the model on the STL-10 dataset. The latent functions in $G$-CNN are defined on $\mathbb{Z} \rtimes G$, where $G$ is the rotation or dihedral group. To visualize the equivariance error, we pool the latent function over the group $G$ following the method in Cohen & Welling (2016), ensuring equivariance with respect to the group action on the input $\rho_g$. For an input image $I$, layer $k$, and equivariant $G$-CNN $\mathcal{G}$, we defined the equivarinace error at each spatial location $(i, j)$ as

$$E_{eq}^k[i,j] = \frac{\left(\mathtt{Pool_g}\mathcal{G}^k(I)[i,j] - \left(\rho_{g^{-1}}\mathtt{Pool_g}\mathcal{G}^k(\rho_g I)\right)[i,j]\right)^2}{||\mathtt{Pool_g}\mathcal{G}^k(I)||_2^2}, \tag{A28}$$

where $\mathcal{G}^k(I)$ denotes the output of $k^{th}$ hidden layer given the input $I$ and $g \in G$. In other words, the equivariant error $E_{eq}^k[i,j]$ denotes the equivariance error at pixel $(i,j)$. As can be observed in Fig. A1-A8, our approach with anti-aliasing shows lower equivariance error throughout the features at all three layers. Recall, that our experiment uses an architecture consisting of a downsampling operation between each of the layers. We also observe that the equivariance error indeed propagates and worsens deeper into the models. Finally, we note that perfect equivariance, i.e., zero error, is not achieved due to the boundary pixels going out of scope in the rotated images.

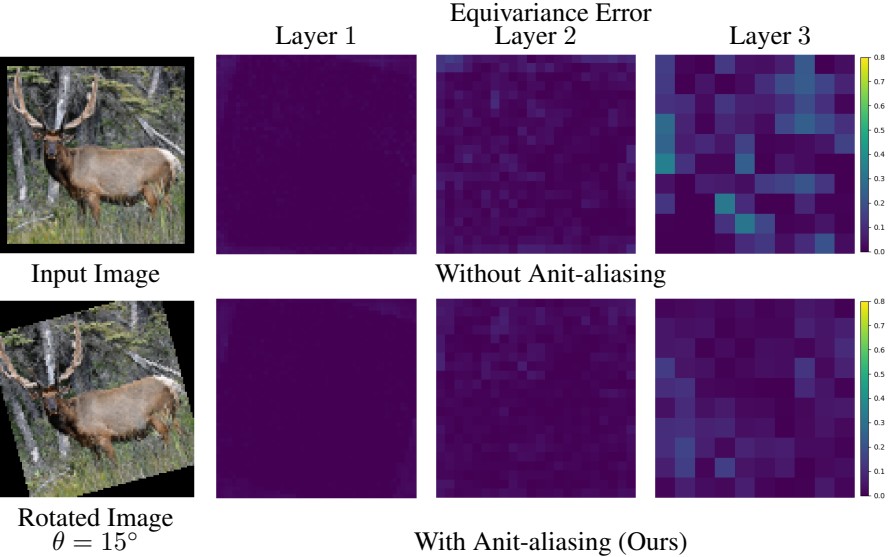

Figure A1: Visualization of the equivariance error at each layer.

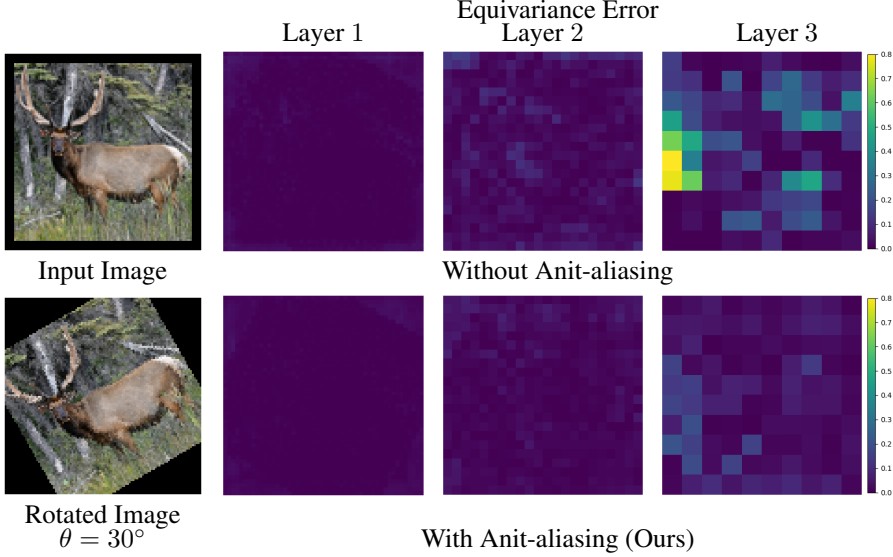

Figure A2: Visualization of the equivariance error at each layer.

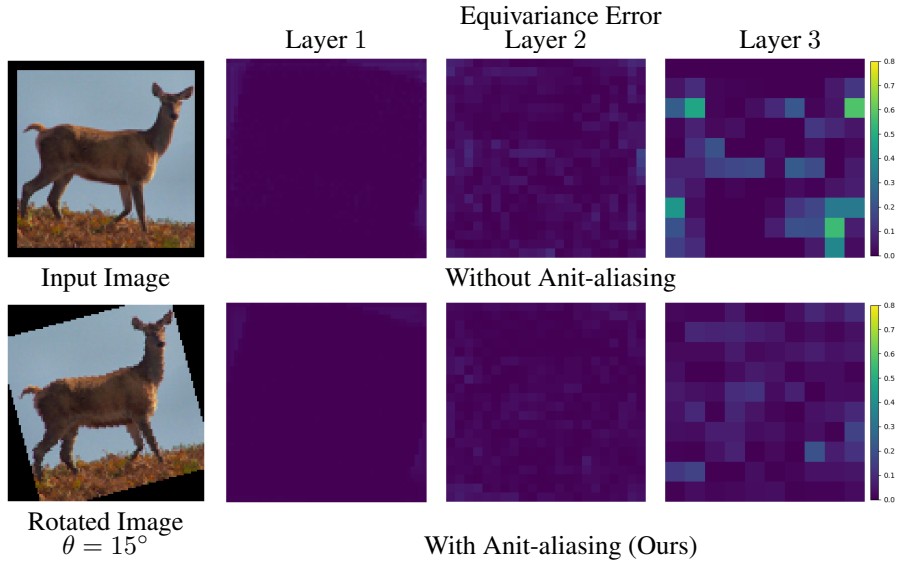

Figure A3: Visualization of the equivariance error at each layer.

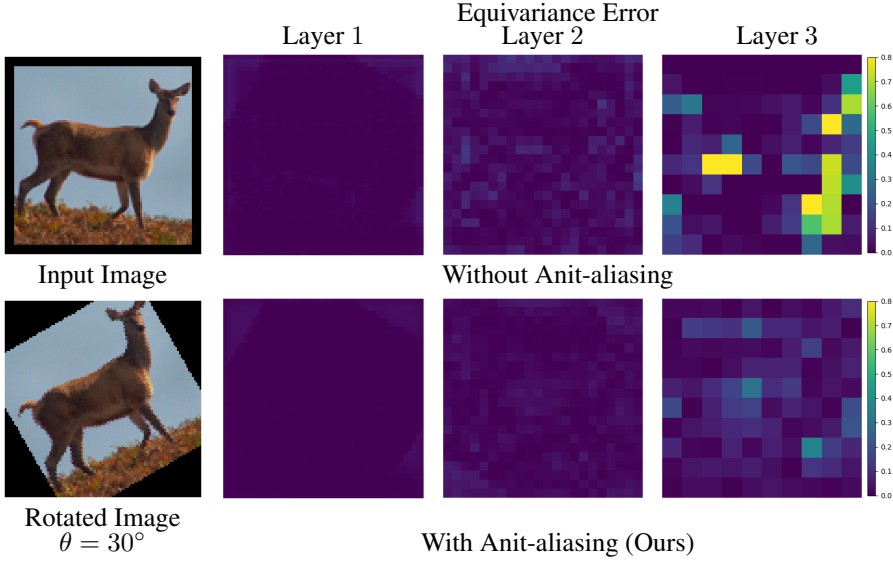

Figure A4: Visualization of the equivariance error at each layer.

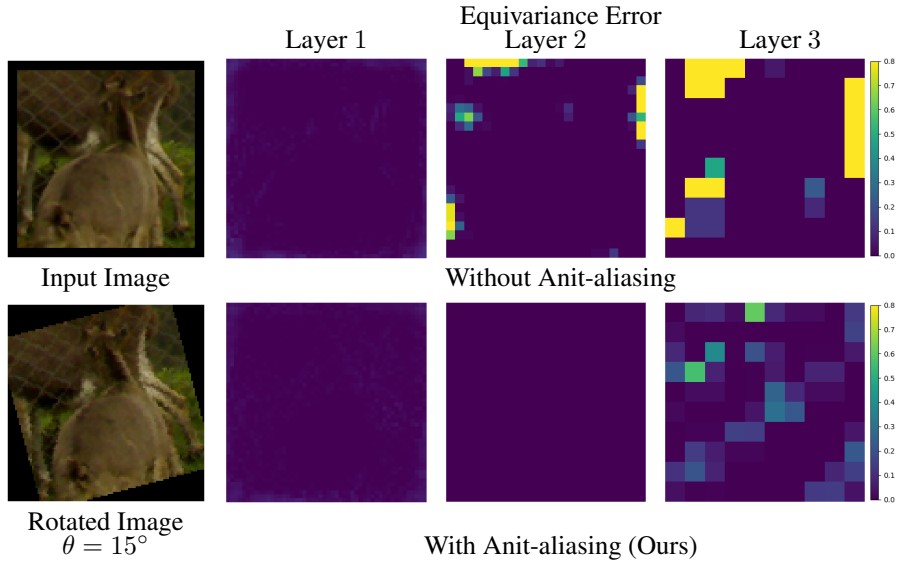

Figure A5: Visualization of the equivariance error at each layer.

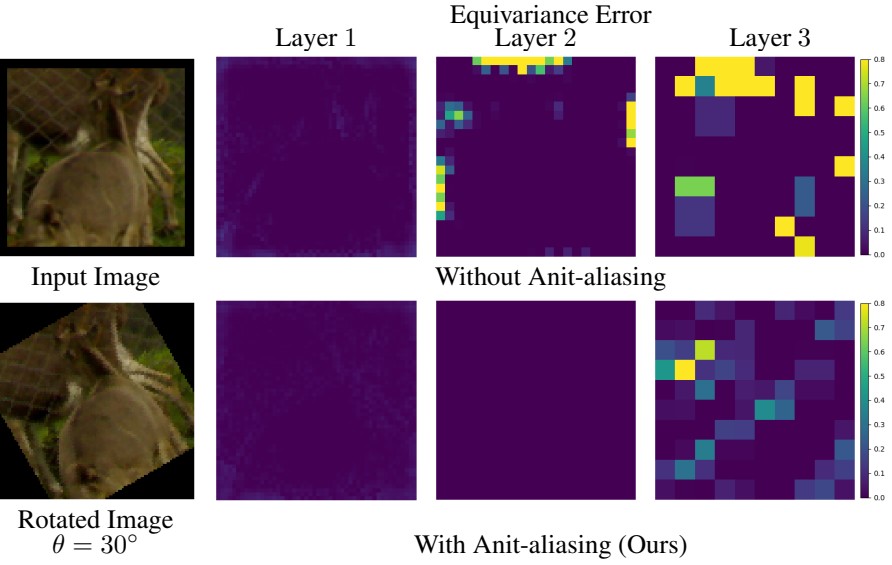

Figure A6: Visualization of the equivariance error at each layer.

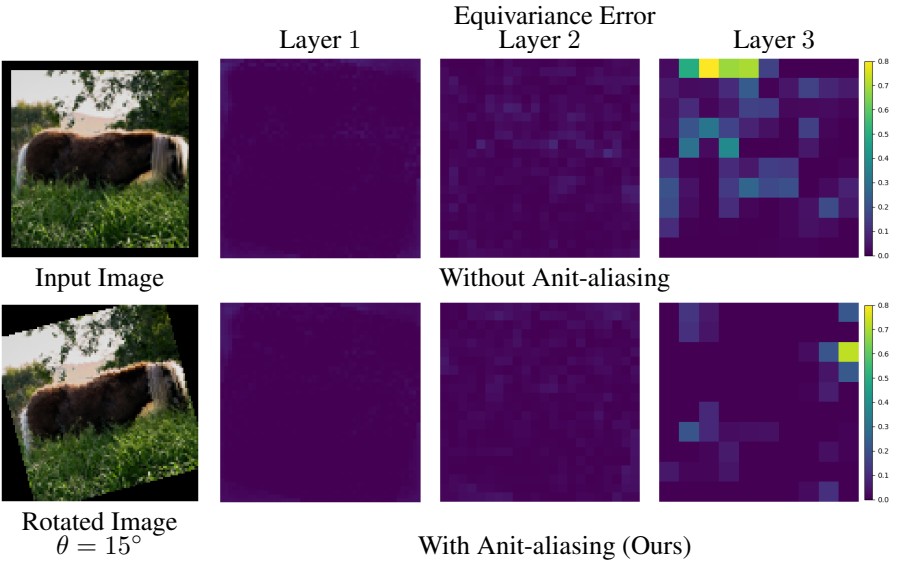

Figure A7: Visualization of the equivariance error at each layer.

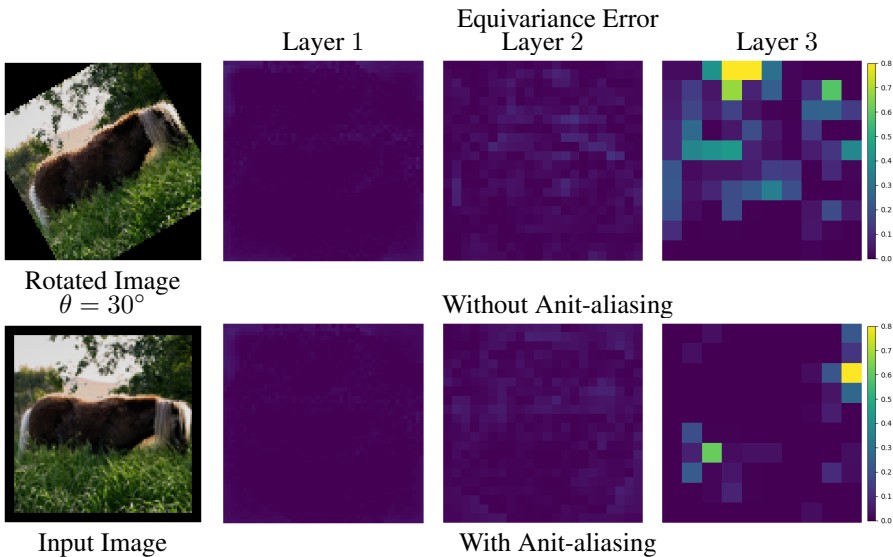

Figure A8: Visualization of the equivariance error at each layer.

## A3 COMPLETE PROOFS OF LEMMAS AND CLAIMS

### A3.1 PROOF OF LEMMA 1

First, we provide the proof of the lemmas.

**Lemma 1.** *For the set $G^{\downarrow}$ returned by Alg. 1, $v \in G^{\downarrow}$ if and only if $v$ can be expressed as a product of the elements of the set $S^{\downarrow} = \left(S/\{s_d\}\right) \cup \{s_d^R\}$.*

*Proof.* In the graph $(V, E')$, there exists a path between $e$ and some node $v \in V$ iff $v \in G^{\downarrow}$ (guaranteed by BFS traversal algorithm). So it will be sufficient to prove that in a graph $(V, E')$, there exists a path between node $e$ and some node $v$ if and only if $v$ can be expressed as the product of elements of $S^{\downarrow}$.

By construction, each node in a directed Cayley graph $(V, E) = \mathtt{DiCay}(G, S)$ has an out-degree of $|S|$, with each outgoing edge corresponding to an element of the set $S$. Removing all outgoing edges corresponding to the element $s_d$ and adding a new outgoing edge to each node corresponding to the new element $s_d^R$, i.e., $E' = E \setminus \{(a, a \cdot s_d) : a \in V\} \cup \{(a, a \cdot s_d^R) : a \in V\}$ maintains the property with respect to $S^{\downarrow}$. That means each node in graph $(V, E')$ has an outgoing edge corresponding to each element of the set $S^{\downarrow}$.

Let's assume there exists a path from $e$ to node $a \in V$. We denote the path as a list of vertices by $\{e, (e \cdot s_{a_1}), \dots, (e \cdot s_{a_1} \dots \cdot s_{a_{m-1}} \cdot s_{a_m})\}$ which is constructed by picking $m$ hops from $e$ along the edges corresponding to the elements $\{s_{a_1}, \dots, s_{a_{m-1}}, s_{a_m}\}$ in order where $\forall j \; s_{a_j} \in S^{\downarrow}$. This implies $a = \prod_{j=1}^{m} s_{aj}$, i.e., $a$ is generated by products of the elements of set $S^{\downarrow}$.

Conversely, let $b = \prod_{i=1}^{n} s_{b_i}$ such that $\forall i \; s_{b_i} \in S^{\downarrow}$. Existence of a path from $e$ to $b$ demands the existence of a series of hops from $e$ along the edges corresponding to the elements $s_{b_1}, \dots, s_{b_{n-1}}, s_{b_m}$. Such a series of hops always exists in graph $(V, E')$ as every node has $|S'|$ out-going edges corresponding to each element in $S'$. $\square$

### A3.2 PROOF OF LEMMA 2

**Lemma 2.** *For the set $S^{\downarrow}$ in Lemma 1, each element $s_i \in S^{\downarrow} \implies s_i^{-1} \in G^{\downarrow}$.*

*Proof.* Let $s_k \in S^{\downarrow} \setminus \{s_d^R\}$, then $s_k^{-1} = s_k^{o_k - 1}$ (as $o_k$ is order of $s_k$), i.e., $s_k^{-1}$ can be expressed as a product of the elements of $S^{\downarrow}$ by Lemma 1, $s_k^{-1} \in G^{\downarrow}$.

Now $(s_d^R)^{-1} = s_d^{-R}$. Let, $w = (o_d - 1)$ and $(Rw \mod o_d) \equiv (Ro_d - R \mod o_d) \equiv (-R \mod o_d)$. so, $s_d^{-R} = s_d^{wR}$. And, following Lemma 1, $(s_d^R)^{-1} \in G^{\downarrow}$. $\square$

### A3.3 PROOF OF CLAIM 1

**Claim 1.** *If $S_d^k = \{s_d^k : k \in \mathbb{Z}^+ \text{ and } k \mod R \not\equiv 0\}$ are non-redundant powers of $s_d$, $o_d \mod r \equiv 0$, and the elements of $S_d^k$ can not be represented as a product of the elements of the left cosets of the subgroup $G_{sub} = \langle S/\{s_d\} \rangle$ generated by the set $\{s_d^{nR} : n \in \mathbb{Z}_0^+\}$ then Alg. 1 returns a proper subgroup $G^{\downarrow} \subset G$.*

*Proof.* We first prove that $G^{\downarrow}$ is a group.

**Existence of Identity** By construction, $e$ is always a member of set $G^{\downarrow}$ as we start the traversing the graph from node $e$.

**Closure** Let $a, b \in G^{\downarrow}$. Therefore, by Lemma 1 $a = \prod_{j=1}^{m} s_{a_j}, b = \prod_{i=1}^{n} s_{b_i}$ with $\forall s_{a_i}, s_{b_j} \in S^{\downarrow}$. Now $a \cdot b = (\prod_{i=1}^{m} s_{a_i}) \cdot (\prod_{j=1}^{n} s_{b_j})$, i.e, $a \cdot b$ can also be expressed as a product of elements of $S^{\downarrow}$. So, by Lemma 1, $a \cdot b \in G^{\downarrow}$.

**Associativity** As $G^{\downarrow} \subseteq G$, and element of $G^{\downarrow}$ follows the multiplication table of group $G$. So, the associativity of $\langle \cdot \rangle$ operation will hold trivially for elements of $G^{\downarrow}$.

**Existence Inverse element** Let, $v \in G^{\downarrow}$ and $v = \prod_{i=1}^{n} s_{v_i} = s_{v_1} \cdot s_{v_2} \ldots \cdot s_{v_n}$. Now we construct a group element $u$ as $u = s_{v_n}^{-1} \cdot s_{v_{n-1}}^{-1} \ldots s_{v_1}^{-1}$. And, we can see that $v \cdot u = u \cdot v = e$. So, $u = v^{-1}$. By Lemma 2, $\forall i \ s_{v_i}^{-1} \in G^{\downarrow}$ and following the Closure property $u \in G^{\downarrow}$, *i.e.*, $G^{\downarrow}$ is a group.

Now, we prove that $G^{\downarrow} \subset G$ by contradiction. We assume that $\exists s_d^{k_i} \in S_d^k$ such that $s_d^{k_i} \in G^{\downarrow}$.

As the elements of $S_d^k$ are non-redundant, $s_d^{k_i}$ can not be generated only by the generator $S'^{\downarrow} = S^{\downarrow}/\{s_d^R\}$. Additionally, $k_i \mod R \not\equiv 0$ and $o_d = wR$ for some $w \in \mathbb{Z}$ ( $R$ divides $o_d$), $\nexists l \in \mathbb{Z} : lR \mod o_d \equiv k_i$. So, $\nexists l \in \mathbb{Z}$ such that $(s_d^R)^l = s_d^{k_i}$. Therefore, the generators for the element $s_d^{k_i}$ must include $s_d^R$ and elements from the set $S'^{\downarrow}$.

Without any loss of the generality, assume that the path from $e$ to $s_d^{k_i}$ is the shortest among the elements of $S_d^k \cap G^{\downarrow}$. Lets $s_d^{k_i} = s_{k_1} \cdot s_{k_2} \cdot s_{k_2} \ldots s_{k_{n-1}} \cdot s_{k_n}$ such that $\forall i \ s_{k_i} \in S^{\downarrow}$. Now, $s_{k_1}$ can not be $s_d^R$. As

$$s_d^{k_i} = s_d^R \cdot s_{k_2} \cdot s_{k_2} \ldots s_{k_{n-1}} \cdot s_{k_n} \quad \Longrightarrow \quad s_d^{k_i - R} = s_{k_2} \cdot s_{k_2} \ldots s_{k_{n-1}} \cdot s_{k_n},$$

where $k_i - R \mod R \not\equiv 0$ as $k_i \mod r \not\equiv 0$. But $s_d^{k_i-R} \in S_d^k$ requires one less generator, thus contradicting our assumption that the path from $e$ to $s_d^{k_i}$ is the shortest among the elements of $S_d^k \cap G^{\downarrow}$.

A similar restriction is also applicable for $s_{k_n}$. So, the path from $e$ to $s_d^{k_i}$ must start and end with generators from set $S'^{\downarrow}$. Therefore, we can express $s_d^{k_i}$ as

$$s_d^{k_i} = q_1 \cdot (s_d^{Rn_2} \cdot q_2) \cdot (s_d^{Rn_3} \cdot q_3) \ldots \cdot (s_d^{Rn_l} \cdot q_l) \tag{A29}$$

where, $\forall j \ q_j \in G_{sub}$ with $G_{sub}$ is subgroup generated by $S'^{\downarrow}$, and $\forall i \ n_i \in \mathbb{Z}$.

Next, $s_d^{Rn_m} \cdot q_m$ for $2 \le m \le l$ is an element of the left coset of the subgroup $G_{sub}$ generated by element $s_d^{Rn_m}$, i.e., $s_d^{Rn_m} \cdot q_m \in \{s_d^{Rn_m} \cdot g : g \in G_{sub}\}$ and $q_1$ is an element of a trivial left coset of $G_{sub}$ generated by $e$. Therefore, $s_d^{k_i}$ is expressed as the product of the elements of the left cosets of $G_{sub}$ generated by the set $\{s_d^{nR} : n \in \mathbb{Z}_0^+\}$, which contradicts our assumption.

This means that $s_d^{k_i} \notin G^{\downarrow} \ \forall s_d^{k_i} \in S_d^k$ and implies that $G^{\downarrow} \subset G$. $\qquad \square$

### A3.4 PROOF OF CLAIM 2

**Claim 2. Subgroup Sampling Theorem.** *For any signal* $\mathbf{x}$ *on* $G$, *if the Fourier coefficients* $\hat{\mathbf{x}}$ *are in the 1-eigenspace of* $\bar{\mathcal{M}} \triangleq \mathcal{M}(\mathcal{M}^{\dagger}\mathcal{M})^{-1}\mathcal{M}^{\dagger}$ *then it can be reconstructed perfectly from the subsampled signal* $\mathbf{x}^{\downarrow}$ *on* $G^{\downarrow}$. *The superscript* $\dagger$ *denotes the conjugate transpose.*

*Proof.* First, we show that if $\hat{\mathbf{x}}$ is in the 1-Eigenspace of $\mathcal{M}(\mathcal{M}^{\dagger}\mathcal{M})^{-1}\mathcal{M}^{\dagger}$, then $\mathbf{x} \in \text{Span}(\mathcal{B})$ with $\mathcal{B} \triangleq F_G^{-1}\mathcal{M}$. If $\hat{\mathbf{x}}$ is in the 1-eigenspace, then

$$\hat{\mathbf{x}} = \mathcal{M}(\mathcal{M}^{\dagger}\mathcal{M})^{-1}\mathcal{M}^{\dagger}\hat{\mathbf{x}} \tag{A30}$$

$$\Rightarrow \mathcal{F}_G \mathbf{x} = \mathcal{F}_G \mathcal{F}_G^{-1} \mathcal{M}(\mathcal{M}^{\dagger}\mathcal{M})^{-1}\mathcal{M}^{\dagger}\mathcal{F}_G \mathbf{x} \ (\text{as } \mathcal{F}_G \mathcal{F}_G^{-1} = \mathbf{I}) \tag{A31}$$

$$\Rightarrow \mathbf{x} = \mathcal{F}_G^{-1} \mathcal{M}(\mathcal{M}^{\dagger}\mathcal{M})^{-1}\mathcal{M}^{\dagger}\mathcal{F}_G \mathbf{x} \tag{A32}$$

$$\Rightarrow \mathbf{x} = \mathcal{F}_G^{-1} \mathcal{M}(\mathcal{M}^{\dagger}\mathcal{F}_G^{-1\dagger}\mathcal{F}_G^{-1}\mathcal{M})^{-1}\mathcal{M}^{\dagger}\mathcal{F}_G^{-1\dagger}\mathbf{x} \ (\text{as } \mathcal{F}_G^{-1\dagger} = \mathcal{F}_G) \tag{A33}$$

$$\Rightarrow \mathbf{x} = \mathcal{F}_G^{-1} \mathcal{M}((\mathcal{F}_G^{-1}\mathcal{M})^{\dagger}\mathcal{F}_G^{-1}\mathcal{M})^{-1}(F_G^{-1}\mathcal{M})^{\dagger}\mathbf{x} \tag{A34}$$

$$\Rightarrow \mathbf{x} = \mathcal{B}(\mathcal{B}^{\dagger}\mathcal{B})^{-1}\mathcal{B}^{\dagger}\mathbf{x} \tag{A35}$$

$$\Rightarrow \mathbf{x} = \mathcal{P}_{\mathcal{M}}\mathbf{x} \tag{A36}$$

Here, $\mathcal{P}_{\mathcal{M}} \triangleq \mathcal{B}(\mathcal{B}^{\dagger}\mathcal{B})^{-1}\mathcal{B}^{\dagger}$ denotes the projection matrix to the column space of $\mathcal{B}$. Note that the columns of $\mathcal{B}$ are linearly independent. As $\mathcal{F}_{G^{\downarrow}}^{-1} = \mathcal{S}\mathcal{B}$ and $\text{rank}(\mathcal{F}_{G^{\downarrow}}^{-1}) = M$. The $\text{rank}(\mathcal{B})$ is at least $M$. And as $\mathcal{B}$ has $M$ columns, they are independent, and $\text{rank}(\mathcal{B}) = M$, $\mathcal{P}_{\mathcal{M}}$ is a valid projection matrix.

This means that $\mathbf{x}$ is in $\text{Span}(\mathcal{B})$, *i.e.*, we can express $\mathbf{x} = \mathcal{B}\hat{\mathbf{x}}_c$ for some set of coefficient vector $\hat{\mathbf{x}}_c$. Perfect reconstruction from the subsampled signal $\mathbf{x}^{\downarrow}$ is now possible, *i.e.*,

$$\mathcal{I}\mathbf{x}^{\downarrow} = (\mathcal{B}\mathcal{F}_{G^{\downarrow}})(\mathcal{S}\mathbf{x}) = (\mathcal{B}\mathcal{F}_{G^{\downarrow}}\mathcal{S})(\mathcal{B}\hat{\mathbf{x}}_c) = \mathcal{B}\mathcal{F}_{G^{\downarrow}}\mathcal{F}_{G^{\downarrow}}^{-1}\hat{\mathbf{x}}_c = \mathcal{B}\hat{\mathbf{x}}_c = \mathbf{x}. \tag{A37}$$

In conclusion, perfect reconstruction of $\mathbf{x}$ is possible from $\mathbf{x}^{\downarrow}$ when $\hat{x}$ is in the 1-Eigenspace of $\mathcal{M}(\mathcal{M}^{\dagger}\mathcal{M})^{-1}\mathcal{M}^{\dagger}$.

$\square$

## A4    ILLUSTRATION OF CLAIM 1

Here is an illustration of Claim 1. The claim states that if $S_d^k = \{s_d^k : k \in \mathbb{Z}^+ \text{ and } k \mod r \not\equiv 0\}$ are non-redundant powers of $s_d$, $o_d \mod r \equiv 0$, and elements of $S_d^k$ can not be represented as a product of elements of left cosets of the subgroup $G_{sub} = \langle S/\{s_d\} \rangle$ generated by set $\{s_d^{nR} : n \in \mathbb{Z}_0^+\}$ then Alg. 1 returns a proper subgroup $G^\downarrow \subset G$. In Fig. A9, we illustrate the claim with group $D_8$.

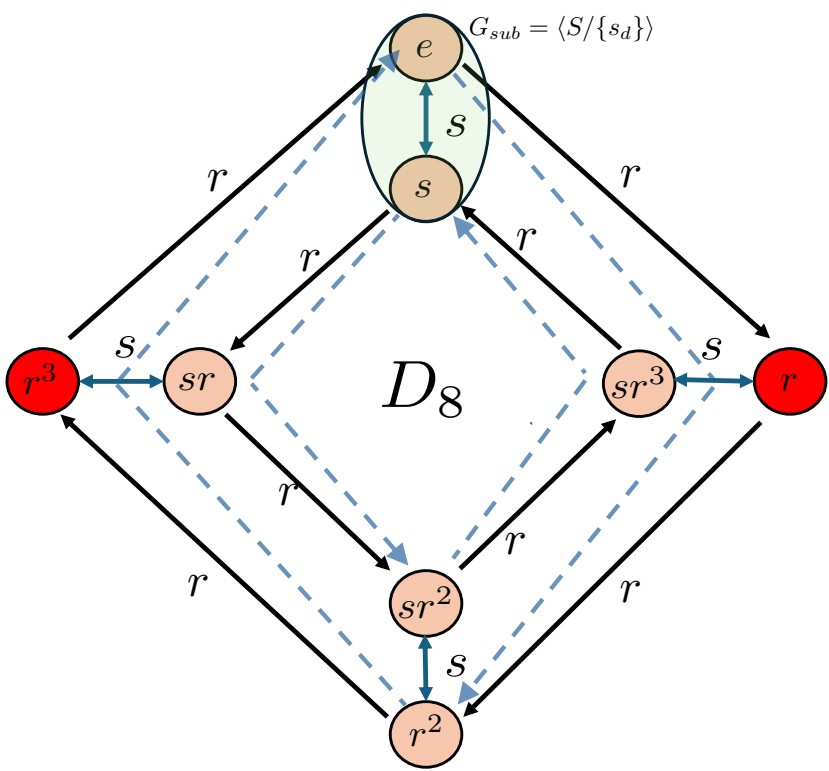

Figure A9: Illustraion of Claim 1 for subsampling $D_8$ by a factor $R = 2$ along the generator $s_d = r$. The red-colored nodes denote the set $S_d^k = \{r, r^3\}$. The green highlighted nodes $\{e, s\}$ is the $G_{sub} = \langle S/r \rangle$. We can see $S_d^k$ is nonredundant, and the order of $s_d$ is divided by 2. The last part of the claim implies that the colored node must not be reachable from nodes $G_{sub}$ with a hop of $r^2$ denoted in a dotted blue line. Which is indeed satisfied with the example shown.

## A5    GENERALIZATION OF SAMPLING ALGORITHM

In this section, we provide an algorithm (see Alg. 2) to check for compliance of a generator with the condition in Claim 1. We also provide a general sampling algorithm (Alg. 3) that maximizes the number of generators in the subgroup following the heuristics from §4.1.

The Alg. 2 takes $O(|V|+|E|)$ time where $V$ is the set of nodes and $E$ is the set of edges in the Cayley graph. To choose the generator with the highest order, we need to check for compliance for each of the generators, making the time complexity to downsample by a chosen generator $O(|S|.(|V|+|E|))$. The computational complexity can be high for complex groups depending on the choice of the generating set $S$. Since the sampling algorithm runs only once before training to generate the sampling matrix, efficiency is maintained. Furthermore, for large complex groups, such as symmetry groups $S_n$, the subgroups can be selected based on prior domain knowledge followed by our proposed anti-aliasing operation.

---

**Algorithm 2** Check-Compliance

---

1: **Input:** Group $G$, Generators $S$, Generator $s$, Order of the generator $o$, subsampling rate $r$
2: **Output:** `True`, `False`
3: **if** $o \mod r \neq 0$ **then**
4:     **Return** `False`
5: **end if**
6: $V, E \leftarrow \texttt{DiCay(G,S)}$
7: **for** each $v \in V$ **do**
8:     $E.remove((v, v \cdot s_d))$
9:     $E.add((v, v \cdot s_d^r))$
10: **end for**
11: // graph traversal from  e
12: $Q \leftarrow \varnothing$
13: $G_{cosets} \leftarrow \varnothing$
14: $Q.enqueue(e)$
15: **while** $Q \neq \varnothing$ **do**
16:     $n \leftarrow Q.dequeue()$
17:     $G_{cosets}.add(n)$
18:     **for** each $(n, m) \in E'$ **do**
19:         **if** $m \notin Q$ **then**
20:             $Q.enqueue(m)$
21:         **end if**
22:     **end for**
23: **end while**
24: **if** $\exists s^k \in G_{cosets}$ such that $k \mod r \neq 0$ **then**
25:     **Return** `False`
26: **end if**
27: **Return** `True`

---

**Algorithm 3** General-Subsample

---

1: **Input:** Group $G$, Generators $S$, Order of the generators $O$, subsampling rate $r$,
2: **Subsampled Group:** $G^{\downarrow}$
3: $V, E \leftarrow \texttt{DiCay(G,S)}$
4: $G^{\downarrow} \leftarrow G.copy()$
5: $R \leftarrow factorize(r)$
6: **for** $i = 1$ to $R.length()$ **do**
7:     $index \leftarrow \texttt{NULL}$
8:     **for** $j = 1$ to $S.length()$ **do**
9:         // check the compliance of $S[j]$ using Alg. 2
10:         **if** check-compilance$(G, S[j], O[j], R[j])$ **then**
11:             **if** $(index = \texttt{NULL}$ OR $O[j] < O[index])$ **then**
12:                 $index \leftarrow j$
13:             **end if**
14:         **end if**
15:     **end for**
16:     **if** $index = \texttt{NULL}$ **then**
17:         **Return** `NULL`
18:     **end if**
19:     // Downsampling using Alg. 1
20:     $G^{\downarrow} \leftarrow \texttt{Downsample}(G, S, R[i], S[index])$
21:     // updating generating set and order
22:     $S[index] \leftarrow S[index]^{R[i]}$
23:     $O[index] \leftarrow O[index]/R[i]$
24: **end for**
25: **Return** $G^{\downarrow}$

---

## A6  FUNCTION ON GROUPS IN EQUIVARIANT CNN FOR IMAGES

**Group Convolution:** In the group equivariant convolution neural network the input image $f : \mathbb{Z}^2 \to \mathbb{R}^k$ (k=1 or 3 depending on whether the image is grayscale or colored) is first lifted to the space of roto-translation or dihedral-translation group ($\mathbb{Z}^2 \rtimes C_N$ or $\mathbb{Z}^2 \rtimes D_N$) by the lifting operation (Cohen & Welling, 2016)

$$[f \star \psi](g) = \sum_{z \in \mathbb{Z}^2} \sum_k f_k(z) \psi_k(g^{-1}z), \forall g \in \mathbb{Z}^2 \rtimes G, \tag{A38}$$

where $k$ is the channel index, *i.e.*, $f_k$ represents $k$ channel of the image, $\psi_k : \mathbb{Z}^2 \to \mathbb{R}$ is $2D$ kernel, and $G$ is either cyclic (rotation) group or dihedral group ($C_N$ or $D_N$) for most computer vision tasks. This transformation lifts the image to the desired group by repeatedly applying the transformed (by the action of the group) filter on the image $f$.

The filter $\psi_k$ is a regular convolution filter, *i.e.*, it is a real-valued function defined on $2D$ grid $\mathbb{Z}^2$. The action of $\mathcal{G}$ on the filter $\psi$ is defined as

$$[\rho_g \psi](z) = \psi(g^{-1}z) \ \ \forall z \in \mathbb{Z}^2. \tag{A39}$$

In other words, the transformed filter $[\rho_g \psi]$ is defined through the action of the group element $g$ on the $z \in \mathbb{Z}^2$, which we directly use in Eq. (A38). In the case of the rotation group, the group element $g$ corresponds to angle $\theta \in [0, 2\pi]$, and the action of the group element $\theta$ on $z = [u, v] \in \mathbb{Z}^2$ is defined as

$$\theta z \triangleq \begin{bmatrix} \cos\theta & -\sin\theta \\ \sin\theta & \cos\theta \end{bmatrix} \begin{bmatrix} u \\ v \end{bmatrix}. \tag{A40}$$

This is the underlying mechanism of rotating real values function on $2D$ grid (for details, please see (Cohen & Welling, 2016; 2017)).

Next, in the group convolution network, the function $[f \star \psi] : \mathbb{Z}^2 \rtimes G \to \mathbb{R}$ is a function over $\mathbb{Z}^2 \times G$ and is passed on to the following group convolution layers. For the ease of notation, let denote the real-valued function on group $\mathbb{Z}^2 \rtimes G$ as $\mathcal{E}$, i.e., $\mathcal{E} : \mathbb{Z} \rtimes G \to \mathbb{R}$. For any $\mathcal{E}$, the group convolution (Cohen & Welling, 2016) is defined as

$$[\mathcal{E} \star \kappa](g) = \sum_{h \in \mathbb{Z}^2 \rtimes G} \mathcal{E}(h)\kappa(g^{-1}h), \ \ \forall g \in \mathbb{Z}^2 \rtimes G, \tag{A41}$$

where $\kappa : \mathbb{Z} \rtimes G \to \mathbb{R}$ is the group convolution kernel. The output is then passed through point-wise non-linearity and followed by more group convolution layers.

We can see that group convolution is defined by the action of $\mathbb{Z}^2 \times G$ on the function $\kappa$. This is analogous to regular convolution, where the function $\kappa$ is also "shifted" (transformed) by the action of group elements. For example, in the specific case of roto-translation group $\mathbb{Z}^2 \rtimes C_N$, group convolution is analogous to $3D$ convolution where the action of the roto-translation group guides "shift" on the filter. Please see Sec. 7 of Cohen & Welling (2016) and Bekkers et al. (2018) for details.

When performing group convolution, we follow the techniques introduced by Cohen & Welling (2016; 2017) and do not propose any modification of the group convolution operations (Eq. (A38) and Eq. (A41)) introduced in the earlier works. For a detailed explanation and construction of the group equivariant architecture, we refer the readers to Cohen et al. (2019); Weiler & Cesa (2019); Kondor & Trivedi (2018); Cohen & Welling (2016); Bekkers et al. (2018).

**Anti-aliasing:** The function $\mathcal{E}$ is represented as a tensor of size $H \times W \times |G|$, where $H \times W$ is the resolution of the input image, which corresponds to the size of the translation group and $|G|$ is the number of elements in the group $G$.

Now, at a fixed translation group element (spacial location) $(i, j) \in \{0, .., H\} \times \{0, ..., W\}$ of the tensor, we have function over $G$, i.e., $\forall (i, j) \in \{0, .., H\} \times \{0, ..., W\}$, we have

$$\mathcal{E}(h = i, w = j, d) = E_{i,j}(d) \in \mathbb{R} \ \ \forall d \in G, \tag{A42}$$

with $E_{i,j} : G \to R$. This function $E_{i,j}$ is transformed according to the regular representation of $G$. We represent $E_{i,j}$ as a vector of size $|G|$, i.e., $E_{i,j} \in \mathbb{R}^{|G|}$.

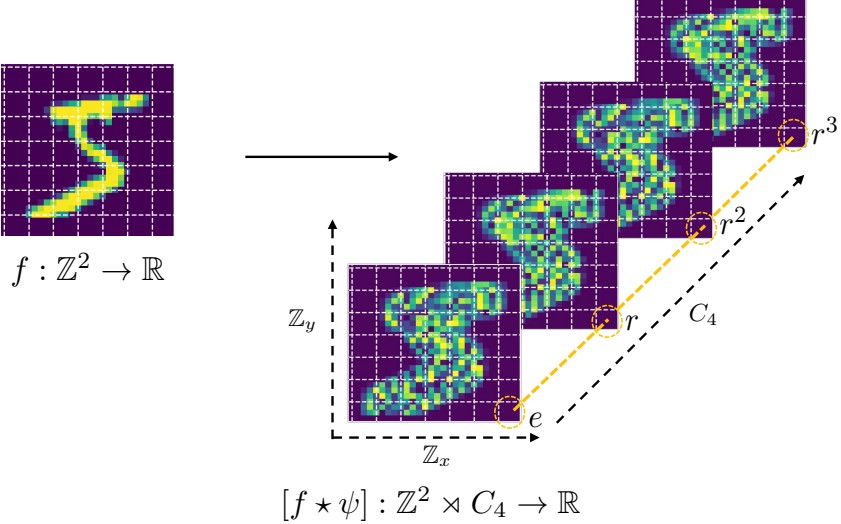

Figure A10: Visualization of function on the group in $C_4 = \{e, r, r^2, r^3\}$ equivariant CNN. The input image $f$ is transformed into a function over a group following Eq. (A38) with some learnable filter $\psi$. The resultant function $[f \star \psi]$ is a function over $\mathbb{Z}^2 \rtimes C_4$. Now at every fixed spatial location $(i, j) \in \mathbb{Z}_x \times \mathbb{Z}_y$, we have functions over $C_4$. Elements of one of such functions are marked with a dotted circle with corresponding elements of $C_4$

In our work, we perform subsampling and anti-aliasing on the functions $E_{i,j}$ $\forall (i, j) \in \{0, .., H\} \times \{0, ..., W\}$. The anti-aliasing operator $\mathcal{P}_{\mathcal{M}*}$ can also be represented as a matrix of size $|G| \times |G|$, *i.e.*, $\mathcal{P}_{\mathcal{M}*} \in \mathbb{R}^{|G| \times |G|}$ and $\mathcal{P}_{gM*}(g, g') \in \mathbb{R}$ $\forall g, g' \in G$. Specifically, the anti-aliasing operation on the function $\mathcal{E}$ is defined as

$$\mathcal{E}'(h = i, w = j, d) = \sum_{l \in G} \mathcal{E}(h = i, w = j, l) \cdot \mathcal{P}_{\mathcal{M}*}(d, l) \ \forall (i, j) \tag{A43}$$

with $\mathcal{E}'(h = i, w = j, d) = E'_{i,j}(d)$.

Finally, Eq. (A43) can be implemented as a matrix-vector multiplication as

$$\mathcal{E}'_{i,j} = \mathcal{P}_{\mathcal{M}*} \mathcal{E}_{i,j}. \tag{A44}$$

In other words, the anti-aliasing is a matrix multiplication along the group dimension of the tensor representation of $\mathcal{E}$.

## A7 ADDITIONAL IMPLEMENTATION DETAILS

It is essential to note that our experiment setup is different from that of Weiler & Cesa (2019) and designed carefully to highlight the robustness of the group equivariant model in a limited data setting. The evaluation metrics are designed to explicitly measure the consistency of the models under all group actions. Unlike Weiler & Cesa (2019), we do not train the randomly rotated datasets, thereby revealing the actual equivariance property of the model by the architecture design. Also, our designed accuracy metrics, $\mathtt{ACC_{orbit}}$, $\mathtt{ACC_{loc}}$, and $\mathtt{ACC_{noaug}}$ provide the performance of the model at different granularity under group actions, which can not be obtained by testing the model on randomly rotated tested (Weiler & Cesa, 2019).

For MNIST, we train on $5,000$ training images without any data augmentation and test on $10,000$ images on different levels of transformations. For CIFAR-10, we train on $60K$ images without any data augmentation and evaluate on $10K$ images. All models consist of 3 group equivariant convolution layers (Cesa et al., 2021; Cohen & Welling, 2016) followed by a linear layer mapping to the final logits. The filter size at each layer is 5. When subgroup subsampling is performed, the convolution layer following the subsampling layer is equivariant only to the subgroup. The output of

the final convolution layer undergoes global-pooling operation (Weiler et al., 2018) to obtain invariant features. For subsampling, roto(dihedral)-translation group, we subsample rotation (dihedral) group and translation group independently. Subsampling along the translation group is equivalent to spatial subsampling and is performed using *BlurPool* (Zhang, 2019). We set $\lambda = 5$ in Eq. (14) for obtaining $\mathcal{M}^*$.

Models are optimized using the Adam optimizer and trained using 15 and 50 epochs with batch sizes of 128 and 256 for MNIST and CIFAR-10 datasets, respectively. All the expenses are run on a single NVIDIA RTX 6000 GPU.

## A8    LATENT FEATURE RECONSTRUCTION

To further investigate the effect of our anti-aliasing operator, we reconstruct the feature on the whole group from the downsampled features on the subgroup. It is crucial to note that the naive subsampling operation in previous work (Xu et al., 2021) lacks a suitable interpolation operation. Hence, we used our proposed interpolation operator to reconstruct the feature, from the first group convolution layer, for both with and without anti-aliasing. We visualize the squared error at each pixel. As shown in Fig. A11, our anti-aliasing operation enables us to reconstruct the original feature across the entire group accurately.

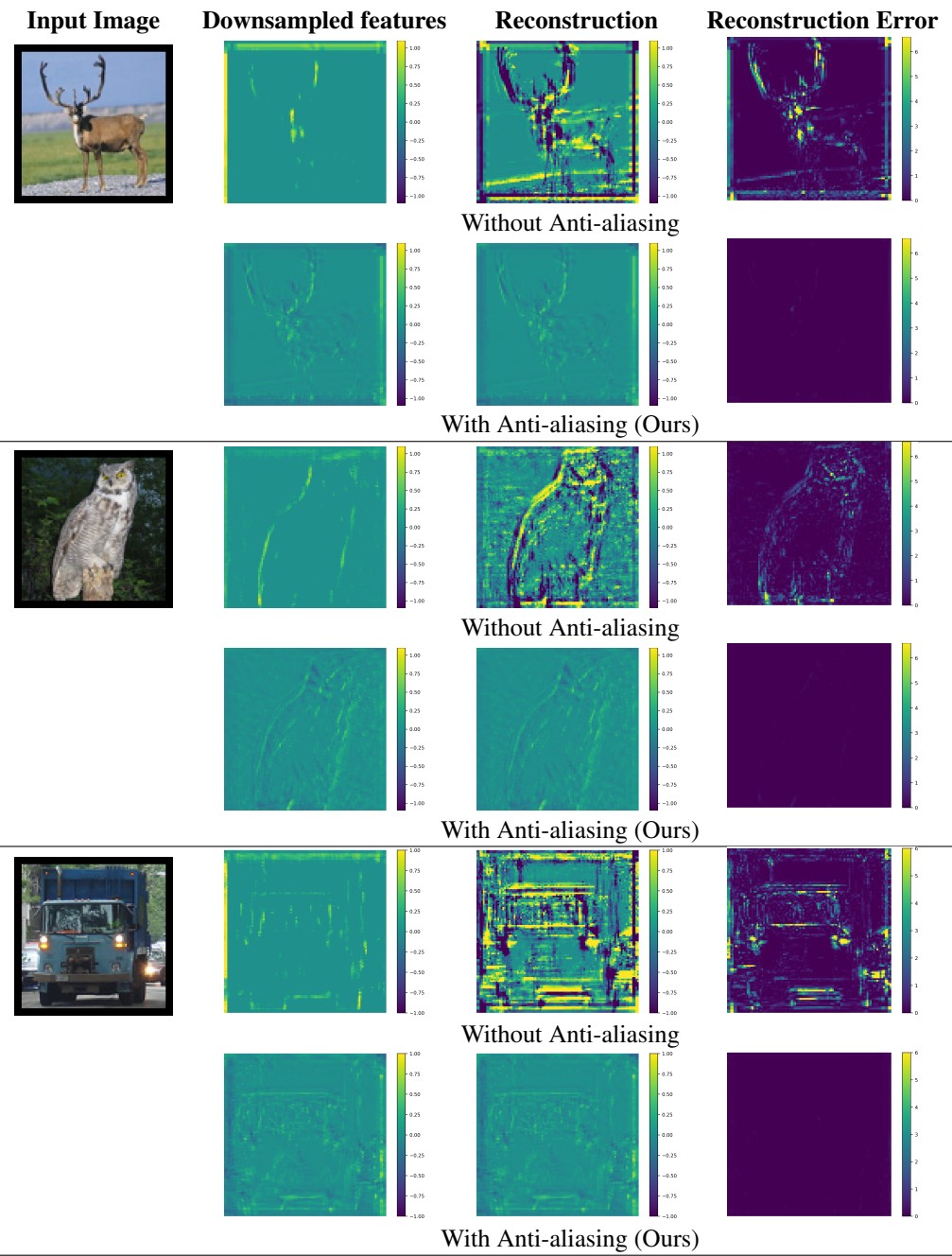

Figure A11: Visualization of the reconstruction quality of the latent feature after subgroup subsampling operation.

