# OpenReview forum: "Group Downsampling with Equivariant Anti-aliasing"
_ICLR.cc/2025/Conference — ICLR 2025 Poster_

### Official Review · Reviewer_rHa3 · 2024-11-01

**Soundness:** 3
**Presentation:** 3
**Contribution:** 2
**Rating:** 6
**Confidence:** 3

**Summary:**

This paper proposes a method for achieving equivariant (finite) group downsampling with anti-aliasing. The main contributions are twofold:
1. Given a downsampling rate, the authors introduce an algorithm to identify an appropriate subgroup within the given finite group.
2. The authors generalize the concept of a low-pass filter to signals over the group, enabling anti-aliasing and perfect reconstruction after downsampling.

Experiments demonstrate that:
1. The anti-aliasing low-pass filter achieves perfect reconstruction post-downsampling.
2. Performance on image classification improves with rotated MNIST and CIFAR-10

**Strengths:**

1. The paper is well-written, well-organized, and relatively easy to follow.
2. The concept of automatically determining a subgroup for downsampling the original group signal, given a specific downsampling rate, is intriguing.
3. The approach of identifying a smooth and equivariant low-pass filter (Eq. 15) before performing group downsampling is also interesting.

**Weaknesses:**

1. A main weakness (as pointed out by the authors themselves) are the limited experiments. The results reported are no where close to the readily-available benchmark on rotated MNIST [1].
2. The authors argued that, in the MNIST experiment, the proposed equivariant-downsampling achieves comparable results with significantly smaller model size. However, one can argue that the original group C24 and D24 are too "large" and probably unnecessary to begin with. It would be interesting to see the comparison to a G-CNN without any subgroup downsampling and start with the smallest group used in the last layer of the proposed method.
3. Also, the advantage of algrithm 1 (automatically determine a subgroup based on the subsampling rate) on Cn and Dn is not clear. I would be much interested to see its effectiveness when dealing with larger groups (such as permutation S(n)).
4. Although I am no expert on the subject, but I would be surprised that perfect reconstruction for group signals after subsampling on a subgroup has not been considered in the literature. Can the authors cite more results on this subject and compare their result to the available ones?


[1] Weiler, Maurice, and Gabriele Cesa. "General e (2)-equivariant steerable cnns." Advances in neural information processing systems 32 (2019).

**Questions:**

1. After subgroup downsampling, the model will only be equivariant to the subgroup. What is the rationale of imposing an equivariant constraint in Eq. 15 on the original larger group?

---

> ### Author Response · Authors · 2024-11-20
> **Rebuttal**
>
> We thank the reviewer for appreciating our work and providing valuable feedback.
>
> `Q18a. On the difference from the benchmark of [1]`
>
> We thank the reviewer for the opportunity to clarify the difference between the rotated MNIST experiment in [1] and our paper.
>
> [1] Weiler et al., General E(2) - Equivariant Steerable CNNs, NeurIPS 2019
>
> Our experiment aims to measure the models' equivariance guarantee at a finer granularity and evaluate model performance in a limited data setting. Hence, we consider a different experiment setup and use different evaluation metrics. Due to the significant differences in experimental setup and evaluation metrics, a direct comparison with the Tables of [1] is not meaningful.
>
> However, we would like to explain why the experiment is still a meaningful comparison to the method in [1].
>
>
> First, we want to clarify that, in Table 2, the model without any subgroup subsampling is equivalent to the architecture proposed in [1].
>
> In [1], the training set is composed of $12K$ randomly rotated (+reflected) images of MNIST. The test set is also composed of random rotations of the original MNIST test set. This means that the model in [1] is trained on rotated samples and evaluated on similarly rotated data.
>
>
> In practice, equivariant CNNs are more applicable for scenarios with limited data, where regular CNN fails and equivariance properties of $G$-CNN models enable better generalization. To reflect this, our experiment uses a more conservative setup designed to simulate a scenario with data scarcity.
>
> In our experiment, the training set is composed of $5K$ unrotated MNIST images.
>
> We test on
> 1. Unrotated MNIST test set($\tt ACC_{\tt no aug}$)
> 2. All possible rotation (reflection) of MNIST test set ($\tt ACC_{orbit}$)
> 3. Rotated test set with angle less than $60^\circ$ ($\tt ACC_{loc}$).
> We follow the same setup for the CIFAR experiment.
>
> The reasons for such choices are:
> a. Unlike [1], the model does not observe any rotated samples during training, which prevents the model from learning the symmetry constraints from the training data and reveals the model's actual equivariance property by architectural design (i.e., equivariance constraint imposed by design).
>
> b. Our evaluation metrics make it easy for us to evaluate the model's performance specifically under all possible group actions ($\tt ACC_{orbit}$), group action within some limit ($\tt ACC_{loc}$), and without any group action ($\tt ACC_{\tt no aug}$). On the other hand, if we test on a randomly rotated test set (as done in [1]), we can not explicitly measure the actual consistency of the model under all group action.
>
> In the updated manuscript, we explain the rationale behind our experimental setup (Appendix Sec A7 Lines 1190-1197).
>
>
> In order to show the limitations of the experiment setup of [1], we now report the results following their setup. We train our models on $12K$ randomly rotated  (+refleted for $O(2)$ symmetry) training data.
>
> We report the accuracy on a similary augmented test set. We observe that the performance now matches the performance reported in [1] (see Figure 2 [1]). However, as the training set is large and also contains augmented data, all models achieve high accuracy and the gap between them is minimal. Despite the similar in performance, we still observe that anti-aliasing always improved the performance by a small amount.
>
> ## SO(2) Symmetry
>
> | Sub. R. | $\mathcal{P}_{\mathcal{M*}}$ | ACC   |
> |---------|--------------------|---------|
> |         | -                  | 0.9901 |
> | 2       | X                  | 0.9855 |
> | 2       | $\checkmark$       | 0.9902 |
> | 3       | X                  | 0.9863 |
> | 3       | $\checkmark$       | 0.9898 |
> | 4       | X.                 | 0.9880 |
> | 4       | $\checkmark$       | 0.9888 |
>
>
> ## O(2) Symmetry
>
> | Sub. R. | $\mathcal{P}_{\mathcal{M*}}$ | ACC    |
> |---------|------------------------------|--------|
> |         | -                            | 0.9830 |
> | 2       | X                            | 0.9852 |
> | 2       | $\checkmark$                 | 0.9877 |
> | 3       | X                            | 0.9851 |
> | 3       | $\checkmark$                 | 0.9852 |
> | 4       | X                            | 0.9861 |
> | 4       | $\checkmark$                 | 0.9874 |

---

> > ### Author Response · Authors · 2024-11-20
> >
> > `Q18b. On limited experiments`
> >
> > We acknowledge that additional experiments could further validate our approach. As mentioned in the limitation section, we believe our contribution is mainly in the theoretical aspect; this is also reflected in our bullets of contributions. The paper aims to provide a theoretical foundation for signal processing on functions over groups, particularly in group-equivariant architectures in computer vision.
> >
> > The experiments on CIFAR-10 (natural images) and MNIST (handwritten digits) were chosen because they strike a balance between complexity and interpretability, allowing us to effectively demonstrate how our proposed theory improves both accuracy and equivariance in $G$-CNNs. For example, for MNIST, we know digits '6' and '9' are confounding classes under the group of 90-degree rotation.
> >
> > The natural images of CIFAR-10 serve as a more complex benchmark, composed of colored natural images, which contain more complex images with higher-frequency details compared to MNIST, which further validates our approach. These application / datasets have been widely used in previous works while validating newly developed theories related to group equivariant models [1,2,3,4].
> >
> > [1] Cohen & Welling, Group equivariant convolutional networks, ICML 2016
> >
> > [2] Weiler et al., Learning steerable filters for rotation equivariant CNNs, CVPR 2018
> >
> > [3] Weiler et al., General E(2) - Equivariant Steerable CNNs, NeurIPS 2019
> >
> > [4] Wang et al., A general theory of correct, incorrect, and extrinsic equivariance, NeurIPS 2023
> >
> >
> > In addition to demonstrating theoretical contributions, our experiments show that our approach improves accuracy and equivariance while reducing the parameter requirement by $50\%$ all while introducing no additional parameter of the anti-aliasing operator with minimal computation overhead.
> >
> > Our current works serve the purpose of filling the missing theoretical link between group theory and signal processing, which is of significant theoretical interest. The experiments are designed to empirically validate our theory alongside the formal proofs. In future work, we plan to extend our model to larger-scale image datasets which is beyond the scope of this work.
> >
> > We hope this clarifies our experimental choices and demonstrates that they are sufficient for validating our theoretical framework.
> >
> > ---
> > In response to the limitation of experiments, we have conducted additional experiments on the **STL-10** [5] dataset with image size $96\times 96$. We can observe that, in this case, our method also improves performance with reduced parameter count. Especially at a given sampling rate, when our proposed anti-aliasing is performed, we see significant improvement in $\tt ACC_{orbit}$ and $\tt ACC_{loc}$. See the Tables below and the updated Appendix.
> >
> > [5] https://cs.stanford.edu/~acoates/stl10/

---

> > > ### Author Response · Authors · 2024-11-20
> > >
> > > # Results on STL-10
> > >
> > > ## SO(2) Symmerty
> > > | Initial Symmetry Group | Sampling Factor |$\mathcal{P}_{\mathcal{M*}}$| #params     | $\tt ACC_{no aug}$ | $\tt ACC_{orbit}$ | $\tt ACC_{loc}$ | $\mathcal{L}_{\tt equi}$ |
> > > |-------------------------|----------------|----------------------------|-------------|--------------------|-------------------|-----------------|--------------------------|
> > > | $C_{24}$               | -               |                            | 1.3M   | 0.54              | 0.30           | 0.34        | 0.16                 |
> > > | $C_{24}$               | $2$             |    $\checkmark$            |962K    | 0.60              | 0.37           | 0.42        | 0.16                 |
> > > | $C_{24}$               | $2$             |    X                       |962K    | 0.60              | 0.35           | 0.40        |0.17                  |
> > > | $C_{24}$               | $3$             |    $\checkmark$            |831K    | 0.62              | 0.37           | 0.42        | 0.16                 |
> > > | $C_{24}$               | $3$             |    X                       |831K    | 0.60              | 0.34           | 0.38        | 0.18                 |
> > >
> > >
> > >
> > > ## O(2) Symmetry
> > > | Initial Symmetry Group | Sampling Factor |$\mathcal{P}_{\mathcal{M*}}$| #params     | $\tt ACC_{no aug}$ | $\tt ACC_{orbit}$ | $\tt ACC_{loc}$ | $\mathcal{L}_{\tt equi}$ |
> > > |-------------------------|-----------------|---------------------------|-------------|--------------------|-------------------|-----------------|--------------------------|
> > > | $D_{24}$               | -               |                            | 1.3M        | 0.57              | 0.32            | 0.37        | 0.12                 |
> > > | $D_{24}$               | 2               |  $\checkmark$              | 962K        | 0.64              | 0.27           | 0.40         | 0.19                 |
> > > | $D_{24}$               | 2               |    X                       | 962K        | 0.61              | 0.26           | 0.40         | 0.20                 |
> > > | $D_{24}$               | 3               |   $\checkmark$             | 831K        | 0.64              | 0.39           | 0.44         | 0.17                 |
> > > | $D_{24}$               | 3               |    X                       | 831K        | 0.60              | 0.33           | 0.33         | 0.17                 |

---

> > > > ### Author Response · Authors · 2024-11-20
> > > >
> > > > `Q19. Justification for using large group`
> > > >
> > > > We thank the reviewer for letting us clarify the motivation behind the experiments on the equivariant neural networks.
> > > >
> > > > We want to emphasize that this paper is primarily theoretical in nature. The primary motivation of our paper is to introduce a sampling theorem and anti-aliasing operator for general discrete groups, an area that has not been explored in the current literature. Our experiments aim to empirically validate our proposed anti-aliasing operator, which is built on the novel Sampling theorem.
> > > >
> > > > **Experimental Justification:**
> > > >
> > > > We understand the reviewer's suggestion to compare our method with a $G$-CNN that starts with smaller symmetry groups (e.g., $D_6$ or $C_6$) without subgroup downsampling and check whether starting from a larger group such as $C_{24}$ or $D_{24}$ is indeed necessary.
> > > >
> > > > The reason for starting with larger groups is two folds:
> > > >
> > > > 1. *Better Consistency Under Group Action*: In the following table, we provide the result of equivariant CNN with symmetry groups equal to the last layer of the sub-sampled g-CNN on MNIST. We observe that even though they achieve a comparable result on the original test set ($\tt ACC_{no\ aug}$), their performance drops under group action ($\tt ACC_{loc}$ and $\tt ACC_{orbit}$) and perform worse than the equivariant model on a larger group such as $C_{24}$ or $D_{24}$.
> > > >
> > > > | Symmetry Group | $\tt ACC_{no\ aug}$ | $\tt ACC_{loc}$ | $\tt ACC_{orbit}$ |
> > > > |----------------|----------------------|-----------------|------------------|
> > > > | $D_6$          | 0.9783             | 0.6912         | 0.4896           |
> > > > | $D_{8}$        | 0.9780             | 0.7470         | 0.6163           |
> > > > | $D_{12}$       | 0.9730             | 0.8128         | 0.8248           |
> > > > | $D_{24}$       | 0.9752              | 0.8253          | 0.8496          |
> > > > | $C_6$          | 0.9772             | 0.6671         | 0.5176           |
> > > > | $C_8$          | 0.9763             | 0.8138         | 0.8241           |
> > > > | $C_{12}$       | 0.9790             | 0.8044         | 0.8237           |
> > > > | $C_{24}$       | 0.9767              | 0.8234          | 0.8346           |
> > > >
> > > > So, the question of whether it is necessary to begin with a larger group depends on what degree of equilibrium or robustness under group action the application requires.
> > > >
> > > > 2. *Demonstration of the method's effectiveness at different downsampled scales*:
> > > > Starting with a large group allows us to demonstrate the effect of sampling at different rates, such as 2, 3, and 4. Thus, it allows us to demonstrate the effect of our proposed method at different scales. However, such a demonstration would not be possible if we started with a small group; there are not enough elements to downsampled from.

---

> > > > > ### Author Response · Authors · 2024-11-20
> > > > >
> > > > > `Q20. On the advantage of Algorithm 1`
> > > > >
> > > > >
> > > > > **On the advantage of Algorithm 1**.
> > > > > The advantage of the proposed algorithm is twofold.
> > > > > 1. *Connection Between Sampling Rate and Subgroup:* Algorithm 1 is the first to establish a direct connection between the sampling rate and subgroup selection, allowing for subgroup subsampling at a specified rate. This is a significant theoretical contribution, as it bridges a previously unexplored gap between signal processing and group theory and opens the door to adapting various signal processing techniques to equivariant models.
> > > > >
> > > > > 2. Combined with our proposed heuristics, the algorithm allows for automatic subgroup selection based solely on the desired sampling rate. It removes the need for manual subgroup selection and maximally preserves action associated with generators
> > > > >
> > > > > In our work, we demonstrate point 1 by mathematical proof and point 2 by providing an ablation study (Table 3), where it achieves better performance. Thus, we appropriately demonstrate the advantages and significance of Algorithm 1.
> > > > >
> > > > >
> > > > > **Applicability to Permutation Groups:**
> > > > > We agree with the reviewer that application to larger groups, such as permutation groups, is more attractive.
> > > > >
> > > > > Our subgroup selection mechanism can be applied to permutation group $S_n$. For example, in case of $S_4$ with generating set $S=\{(1,2), (1,2,3,4) \}$ (using cycle notation), at the sampling rate of $2$ -- our algorithm will return a sub-group $G'$ generated by $S'=\{(1,2), (1,3)(2,4)\}$, which is obtained by setting $s_d = (1,2,3,4)$, i.e., downsampling with respect to the generator $(1,2,3,4)$.
> > > > >
> > > > > However, the second half of the proposed pipeline, i.e., the construction of an anti-aliasing operator based on a selected sub-group, requires an equivariant model which uses functions over the underlying group as features.
> > > > >
> > > > > Permutation group equivariant models are mostly used in designing equivariant models for graphs. Note, that permutation equivariant models for graphs do not use functions over the permutation group as their feature. Instead, these models process the input as a set, ensuring equivariance under permutations of nodes or edges in graphs.
> > > > >
> > > > > We hope this clarifies that Alg. 1 can work with any discrete group, but equivariant models for permutation groups may not be directly compatible with our proposed anti-aliasing operator.
> > > > >
> > > > > For future work, we believe it may be interesting to extend our approach to other types of group-equivariant models.

---

> > > > > > ### Author Response · Authors · 2024-11-20
> > > > > >
> > > > > > `Q21. On the existing literature on Subsampling functions in Group`
> > > > > >
> > > > > >
> > > > > >
> > > > > > In hindsight, anti-aliasing for subsampling on a group seems quite intuitive. However, to the best of our knowledge, we are not aware of any existing work discussing anti-aliasing for group equivariant models. While this may seem surprising, it might be expected when looking back at the history of shift-equivariant models. This further demonstrates the novelty and the potential impact of our work.
> > > > > >
> > > > > > Consider CNNs that have been proposed as early as 1980s, and further popularized in 2014, it was not until [1] in 2019 that signal processing techniques (such as proper anti-aliasing) were introduced. Additionally, designing a general map between bases of an arbitrary group and its subgroup is very non-trivial yet essential for developing an anti-aliasing operator.
> > > > > >
> > > > > > [1] Richard Zhang, Making Convolutional Networks Shift-Invariant Again, ICML 2019
> > > > > >
> > > > > > To the best of our knowledge, there is no prior work that discusses the bandlimited-ness, the Sampling theorem, and the construction of an anti-aliasing operator for general discrete finite groups.
> > > > > >
> > > > > > Next, we now discuss how our work differs from prior work [2,3,4] relating to the Sampling theorem:
> > > > > >
> > > > > > [2] Chen et al., Discrete Signal Processing on Graphs: Sampling Theory, IEEE TSP 2015
> > > > > >
> > > > > > [3] Dodson et al., Groups and the sampling theorem, Sampling Theory in Signal and Image Processing, 2007
> > > > > >
> > > > > > [4] McEwen et al., A novel sampling theorem on the rotation group, IEEE Signal Processing Letters, 2015
> > > > > >
> > > > > > We believe that the closest works are the ones that extend the classic sampling theorem. This includes the studying of bandlimited-ness on graphs [2], or the study of sampling theorem on cyclic and abelian groups [3,4]. (See Line, 083 in related works). In this work, we propose a subgroup sampling theorem for a general finite group that guarantees the perfect reconstruction of the signal on the whole group from the subsampled signal on the subgroup. In Example 3 we also demonstrated how the existing Sampling theorem for cyclic groups is a special case of our proposed theorem.
> > > > > >
> > > > > > In response to the reviewer, in the updated manuscript, we now have more classical works in signal processing on groups [5-6]. These works generalize the discrete Fourier transform to discrete groups but lack generalization for all finite discrete groups (see lines 88-92).
> > > > > >
> > > > > > [5] Vaidyanathan & Kirac, Cyclic LTI Systems in Digital Signal Processing, IEEE TSP 1999
> > > > > >
> > > > > > [6] Napolitano & Spooner, Cyclic Spectral Analysis of Continuous-Phase Modulated Signals, IEEE TSP 2001
> > > > > >
> > > > > >
> > > > > > `Q.22 Rantional for imposing equivariant constraint with respect to the larger group  in Eq. 15`
> > > > > >
> > > > > > There are two key reasons behind this choice:
> > > > > >
> > > > > > 1. **Adaptability with existing group equivariant architectures**: Making the anti-aliasing layer equivariant with the larger group allows us to construct an equivariant hybrid layer when concatenated with the $G$-CNN layer, i.e., convolution followed by anti-aliasing, i.e., it can be easily integrated into existing architecture. For example, the equivariant index selection mechanism described in [1] requires the function to be equivariant with respect to the larger group before their proposed subsampling operation and our proposed anti-aliasing later can be directly integrated (See Section A2.1).
> > > > > >
> > > > > > 2. **Wide use cases:** Our proposed subgroup sampling theorem and anti-aliasing operations are general and also applicable for functions over groups outside of the equivariant architectures. The anti-aliasing operation can be performed independently of the subsampling operation. For example, in traditional signal processing, the anti-aliasing operation can be used as a blurring mechanism to remove high-frequency noise. The same is applicable to our proposed method.
> > > > > >
> > > > > > [1] Xu et al., Group equivariant subsampling, NeurIPS 2021

---

> > > > > > > ### Comment · Reviewer_rHa3 · 2024-11-25
> > > > > > >
> > > > > > > I thank the reviewer for the detailed response. My concerns regarding the limited experiments are largely addressed, particularly the explanation of the rationale for "starting with larger groups" and subsequently subsampling into subgroups. While I do not consider the theory presented in the paper to be groundbreaking, it is a solid contribution, and I have raised my rating accordingly.

---

> > > > > > > > ### Author Response · Authors · 2024-11-27
> > > > > > > >
> > > > > > > > We thank the reviewer for appreciating our work.
> > > > > > > >
> > > > > > > > Regards
> > > > > > > > Authors

---

### Official Review · Reviewer_FQDU · 2024-11-03

**Soundness:** 3
**Presentation:** 2
**Contribution:** 2
**Rating:** 5
**Confidence:** 2

**Summary:**

This paper presents a method in the context of group equivariant neural networks, to design a subsampling operator where we can derive the subsampled group (called subgroup) and a corresponding anti-aliasing method.

**Strengths:**

- I really like that some examples are provided.
- I think it's great to clearly illustrate what claims are tested and under which conditions in experiments.
- Results show the interest of the proposed method in the given context.
- Code is provided to allow the replication of the experiments. (I did not try running them)

**Weaknesses:**

My assessment is based on the lack of clarity of the paper. There small bits and pieces that are confusing but more generally concreteness and link to the application is lacking.

- Clarity: As someone not familiar with group equivariant neural networks, it was very difficult for me to imagine what it means to have a signal indexed by elements of a group. I think section 4 would benefit from having a very visual example beforehand that can explain what all of the objects mentioned are in a concrete setting.
- Experiments: it's not clear what exactly "We apply the proposed subgroup selection and anti-aliasing operator to equivariant CNN architectures" means. I don't understand what is implemented in the code unless I read it, if I stick to the paper.
- Nit: when introducing the order of a generator, maybe it's worth specifying that it always exists and point to a reference.
- Nit: "the minimal generating subset" -> it's not clarified beforehand whether there exists only one.
- Nit (notations):
. $V$ is at one point a vector space, then vertices on a graph. This is confusing.
. The downsampling rate is sometimes $r$ sometimes $R$.

**Questions:**

- In practice, is it always easy to obtain the minimal generating subset for a certain concrete group G? Or maybe another to phrase the question is: given a generating subset for a group, how easy is it to determine whether it is minimal?
- In example 1, is $S = \{\delta t\}$ ? and then we simply pick $s_d = \Delta t$?
- What does "reconstructed perfectly" mean in Claim 2?

---

> ### Author Response · Authors · 2024-11-20
> **Rebuttal**
>
> We thank the reviewer for appreciating our work and providing valuable feedback.
>
> `Q12. Providing an illustration to improve clarity`
>
> We thank the reviewer for the great suggestion. We understand that some concepts related to group-equivariant neural networks (g-CNNs) can be difficult to grasp for readers unfamiliar with this area. To address this concern, in the updated manuscript, we added a section clarifying "what does it mean to have a signal index by the group elements in g-CNN architectures" and "how the anti-aliasing performed on these functions inside the equivariant CNN architectures." Specifically, we describe the details regarding how images are converted to signals on the group in equivariant CNNs and how the group element can index a function. An illustration to explain the concept better is also provided. Please see Appendix Sec A6 and Fig A2 in the updated manuscript.
>
> We refer the reader to this section from lines 390-393 of the main text.
>
> We hope these changes address your concerns and improve the paper's readability. We also highly appreciate and are open to any further suggestions on improving the paper's clarity.
>
> `Q13. Clarification on the order of generators and minimal generating set`
>
> We have added a clarification on the existence of the order of generators along with a reference to standard group theory literature (lines 192-194 ).
>
> We have also clarified the concept of the "minimal generating set": In general, there can be more than one minimal generating set. However, in most practical cases, they are equivalent. For example, for any cyclic group $C_n$ generated by the minimal generating set $S=\{r\}$, $S'=\{r^b\}$  is also a minimal generating set when $b$ and $n$ are relatively prime. (line 127-129 and line 740-742)
>
> `Q14. Suggestions on Notations`
>
> Thanks for pointing this out. we now use different notation ($U$) for vector space. And we now represent all downsampling rates by $R$.
>
> `Q15. Checking for minimal generating set `
>
> *Algorithm to check the minimality of a generating set $S$*:
>
> Let $S$ be a generating set of group $G$ and $b \in S$. If we remove all the edges corresponding to the element $b$ from the Cayley graph of $G$, and even after removing these edges, the Cayley graph remains connected, then $b$ is a redundant element, and the generating set $S$ is not minimal.
>
> To prove $S$ is a minimal generating set, we need to do this test $\forall b \in S$.
>
> The algorithm can be expensive if the Cayley graph is very dense, the group size is large, and the number of elements in the generating set is also large. However, in practice, we already know the minimal generating sets of common groups.
>
> `Q16. Clarification on Example 1`
>
> Yes. For the Example 1, $S=\{ \Delta t\}$ and $s_d = \Delta t$. As only one generator exists, we have to fix $s_d = \Delta t$.
>
> `Q17. Clarification on Perfect Reconstruction`
>
>
> We defined "perfect reconstruction" in lines 291-294. It means when the original signal $x$ and reconstructed signal $\mathcal{I} \mathcal{S}x$ matches exaclty, i.e., $| x - \mathcal{I} \mathcal{S}x|_2^2=0$. In other words, we can get back the original signal on the larger group from the subsampled signal on the subgroup.

---

> > ### Comment · Reviewer_FQDU · 2024-11-26
> >
> > I would like to thank the authors for engaging respectfully in the discussion process.
> >
> > I value the addition of Appendix A.6 which allowed me to re-read the paper in a different light.
> > I know understand what indexing a signal by a group element means: if I rephrase it, we basically lift the signal to include the action of a filter transformed by a each element of a group (rather the minimal generating set). Then we have as many representations of the signal as we have elements in the group (rather the minimal generating set).
> >
> > For this reason, if you have a very large minimal generating set, you may want to downsample on it as well.
> >
> > I think what is still lacking is:
> > - what is the filter $\psi$ in this case and its version transformed by the elements of the group?
> > - what would elements of $D_N$ and $C_N$ be in the specific case of images?
> > - when applied within a G-equivariant-CNN what changes from the case of an input signal described in A.6? In other terms I guess that for images the output of the first layer $x$ is of shape: `[batch_size, n_channels, n_group, height, width]`. What computation is then performed on $x$? In the case of CNNs you just use a convolution (and then nonlinearities etc...) that accepts `[B,N,H,W]` as input, but here because we have this extra dimension it's not clear to me.
> > - as a follow-up to that last question, how is the anti-aliasing filter used in particular?

---

> > > ### Author Response · Authors · 2024-11-27
> > > **Response to the comment of FQDU**
> > >
> > > We thank the reviewer for the feedback.
> > >
> > > `C1. Large minimal generating set`
> > >
> > > Thanks for the thoughtful observation. If the minimal generating set is large, we might want to downsample it as well. However, the minimal generating set itself is not a group, and downsampling on a set is not well defined without additional structures.
> > >
> > > If our goal is to reduce the size of the minimal generating set, that is indeed possible through our proposed approach. Considering the following example:
> > >
> > > The minimal generating set of a Dihedral group is $S=\\{r, s\\}$, where $r$ represents the rotation and $s$ represents the reflection. The order of $s$ is $2$, i.e., $s^2 = e$.
> > >
> > > Now, if during downsampling, we choose $s_d = s$, i.e., we choose to downsample along the generator $s$ and set the downsampling rate to $2$. The new generating set will be $S' = S/\\{s\\} \cup \\{s^2\\} = \\{r\\} \cup \\{e\\}$. Now by definition, $e$ is not a part of the generating set. So the size of the new generating set is reduced by $1$. In this way, the size of the minimal generating set can be reduced when the size is large.
> > >
> > > `C2. Clarification`
> > > 1. We now discuss the details of filter $\psi$ and its transformation under the group action (See lines 1417-1430).
> > > 2. We now clarify that $C_N$ and $D_N$ refer to cyclic and dihedral groups (See line 1414).
> > > 3. We now explain the details of group convolution (See lines 1430-1450) along with specific references for a detailed explanation. After the lifting operation, the shape of the tensor is now `batch_size, n_channels, n_group, height, width`. The group convolution operation can be viewed as a $3D$ convolution on the domain `[n_group, height, width]`, however, here the shift of the filter or convolution kernel is defined by the action of the group element. For a detailed explanation, we refer the reader to the seminal works in the field [1,2,3].
> > > 4. We now have further details on the anti-aliasing operation (see lines 1450-1495).
> > >
> > >
> > >
> > > [1] Cohen & Welling, Group equivariant convolutional networks, ICML 2016
> > >
> > > [2] Weiler et al., General E(2) - Equivariant Steerable CNNs, NeurIPS 2019
> > >
> > > [3] Weiler et al., Learning steerable filters for rotation equivariant CNNs, CVPR 2018
> > >
> > > Please let us know if further clarification needed. We greatly appreciate and thank the reviewer for improving this paper!

---

> > > > ### Author Response · Authors · 2024-12-02
> > > >
> > > > We thank the reviewer for taking the time to review our work and providing important suggestions. In terms of the reviewer's last response,
> > > >
> > > > 1. We have addressed the concern with groups with a large generating set and clarified how our approach could handle it.
> > > > 2. In Sec. A6 of the updated manuscript, we have provided further details on convolution filters, group convolution, and the proposed anti-aliasing operator.
> > > >
> > > > Please review our responses and let us know if additional clarification or modification is needed.

---

### Official Review · Reviewer_fTfZ · 2024-11-04

**Soundness:** 3
**Presentation:** 3
**Contribution:** 2
**Rating:** 6
**Confidence:** 4

**Summary:**

This paper proposes a method to subsample signals over groups by a given factor $R$. The authors, motivated, by signal processing, also provide an anti-aliasing operation to ensure bandlimited-ness and also provide a sampling theorem for groups.

**Strengths:**

The paper has strong connections to signal processing and the approach seems natural from the perspective of signal processing. Concepts are explained well and overall the paper is fairly well-written.

**Weaknesses:**

The paper has a few weaknesses, mainly:

- unclear motivation,
- unexplained choices.

**Motivation**

There are two crucial choices that are made for the paper, but they are never discussed or motivated. Specifically:

- Signals are defined over groups. This is highly non-standard in the equivariance literature, and it’s not clear how the methods presented here fit into the existing space of equivariant works. This is greatly illustrated in Section 5.1 and Figure 3, where the groups $C_{16}$ and $D_{16}$ are used essentially to label the nodes and have no effect on the actual signal. For rotationally-equivariant architectures, we fully expect the feature maps themselves to exhibit equivariance, which is not the case in this work. If that is not possible using this framework, this severely limits the applicability.
- Interpolation in (8) is defined on a subsampled signal. This is highly canonical in signal processing, where interpolation can be applied independently of downsampling. It sees this is necessary for the authors’ analysis, however this is not motivated or explained at all.

**Questions:**

On top of the above weaknesses I had some further questions:

- On line 57 the authors claim that the subsampling layer selects suitable subgroups for the tasks. There are a couple of issues with this: first of all, how is “suitable” defined in this context? In what way do the authors support that claim and how is it evaluated in their experiments? Finally, and maybe most importantly, the statement mentions tasks: however, their subsampling algorithm is task-agnostic and performs the subgroup choice without any knowledge about the downstream task. In what way, then, is the subgroup that is selected suitable for the task?
- The definition of the minimal generating set in 125 is weird. The way things stand, since by construction $e\notin S$, it can never hold that $\langle S \rangle = G$, for any $S$.
- There are multiple typos in the footnote in line 323. There is also a typo in 343 (no space between $\mathcal{M}$ and the previous word), and also on 501 $\textrm{Acc}_{\textrm{orbit}}$ appears twice.

---

> ### Author Response · Authors · 2024-11-20
> **Rebuttal**
>
> We thank the reviewer for appreciating our work and providing valuable feedback.
>
> `Q7. On the connection of our approach to existing equivariant architectures`
>
> Thanks for the opportunity to clarify our connection with equivariant neural networks. For simplicity, we explain using a $2D$ image as an input to the $G$-CNN models.
>
> In the group equivariant convolution neural network (g-CNN) the input image $f: \mathbb{Z}^2 \rightarrow \mathbb{R}^k$ (k=1 or 3 depending on whether the image is grayscale or colored) are first lifted to the space of roto-translation or dihedral-translation group ($\mathbb{Z}^2 \rtimes C_N$ or $\mathbb{Z}^2 \rtimes D_N$) by the following lifting operation [1]
>
> $$\[f \star \psi\](g) = \sum_{y \in \mathbb{Z}^2} \sum_k f_k(y) \psi_k(g^{-1}y)~~\forall g \in \mathbb{Z}^2 \rtimes G,$$
> where, $k$ represents the channel index, i.e, $f_k$ represents $k$ channel of the image, $\psi_k: \mathbb{Z}^2 \rightarrow \mathbb{R}$ is $2D$ kernel, and $G$ is either $C_N$ or $D_N.$
>
> Now the resultant function $\[f \star \psi\]: \mathbb{Z}^2 \rtimes G \rightarrow \mathbb{R}$ is a function over a group and is passed on to the following group convolution layer.
>
> And we have $\[f \star \psi\](h,w,d) \in \mathbb{R}$ when $(h,w) \in \mathbb{Z}^2$ and $d \in G$.
>
> This function $\[f \star \psi\]$ is represented as a tensor of size $H \times W \times |G|$, where $H \times W$ is the resolution of the input image, which corresponds to the size of the translation group and $|G|$ is the number of elements in the considered dihedral group.
>
> Now, at a fixed translation group element (spacial location) $(i, j) \in \{0, .., H\} \times \{0, ..., W\}$ of the tensor, we have function over $G$, i.e.,  $\forall (i,j) \in \{0, .., H\} \times \{0, ..., W\}$, we have
>
> $\[f \star \psi\](h=i,w=j,d) = F_{i,j}(d) \in \mathbb{R},$
>
> with $F_{i,j}: G \rightarrow R$. This function $F_{i,j}$ is transformed according to the regular representation of $G$ under the group action $G$. In the case of steerable CNNs [2], such functions are directly obtained via a steerable filter. In our work, we perform subsampling and anti-aliasing on the functions $F_{i,j} ~~\forall (i,j) \in \{0, .., H\} \times \{0, ..., W\}$.
>
> Now, regarding the concerns about random labeling : the function/signals $F_{i,j}$s are generated by applying the transformed filter $\psi_k(g^{-1}y)$ on the input image. It depends on the symmetry of the input image under the group action, which also exhibits equivariance.
>
> Lastly, we want to clarify that *our framework is applicable to any group equivariant models that use functions or regular representation over finite discrete groups. And our method can be applied to any function on the finite groups without any restriction.*
>
> [1] Cohen & Welling, Group equivariant convolutional networks, ICML 2016
>
> [2] Cohen & Welling, Steerable CNNs, ICLR 2017
>
>
>
> `Q8. On the use of Interpolation Operator`
>
> We agree with the reviewer that we can apply interpolation regardless of the subsampling. However, in Sec 4.2 we aim to establish the Sampling theorem, i.e., the condition under which subsampling followed by interpolation results in perfect recovery of the original signal.
>
> We note that interpolation on the subsampled signals has been considered widely while analyzing the perfect reconstruction condition and proposing the Sampling theorem for function on different domains (see Sec 3 [1], see Chapter 5 [2]). Therefore, we directly introduce interpolation operation in that specific context and apply it only to the subsampled signal on the subgroup as part of our analysis (both proof and method).
>
> To address the concern, in the updated manuscript, we provide clarification that we are following the standard construction of "Sampling followed by interpolation" to establish the Sampling theorem for subgroup subsampling (see line 293-294).
>
> We appreciate and are open to any further suggestions for improving the clarity of the work.
>
> [1] Chen et al, Sampling theory for graph signals, ICASSP, 2015
>
> [2] Vetterli et al., Foundations of signal processing, Cambridge University Press, 2014
>
> `Q9. Clarification on the claims regarding subgroup selection (line 57)`
>
> By "suitable" subgroup, we mean the choice of a subgroup that maximizes the performance and the equivariance guarantee of the model. Specifically, our proposed a heuristic to maximize the number of generators in the subgroup, thus ensuring the maximum preservation of the action associated with the generator and consecutively achieving a better equivariance guarantee. In Table 3, we showed that our choice of subgroup achieved better performance, thus evaluating our proposed heuristics for subgroup selection.
>
> In the mentioned statement, the "tasks" refers to the task of image classifications. We have clarified it in the updated manuscript. (See Line 58)

---

> > ### Author Response · Authors · 2024-11-20
> >
> > `Q10. Clarification on minimal Generating set`
> >
> >
> > $S$ is the generating set of $G$ if all the elements of $G$ can be represented a product of elements of $S$. For example, for rotation group $C_4 = \{e, r, r^2, r^3\}$ the generating set is $S = \{r\}$ (r refers to a rotation of $90^\circ$), as all elements of $C_4$ are actually some power of $r$.
> >
> > Please note that $e = r^4$. So, even if $e \notin S$, $e$ will be generated as a product of elements of $S$, thus generating a valid group $G$. This is always possible as every element of a discrete finite group $G$ is of finite order[1], i.e., for any $b \in S \subset G$, there is an integer $o_b$, such that $b^{o_b}=e$. Thus, we can always generate $e$ a the product of elements of $S$.
> >
> > Now another set $S'=\{r, r^2\}$, by definition, is also a generating of $C_4$, as adding extra elements from $G$ never breaks the condition of being a generating set. The condition of a minimal generating set restricts such redundant elements in the generating set.
> >
> > [1] Martin Isaacs, Algebra: a graduate course, American Mathematical Soc. 2009
> >
> > `Q11. Typos`
> >
> > Thanks for pointing out the typos. We have fixed them in the updated manuscript.

---

> > > ### Author Response · Authors · 2024-12-02
> > >
> > > We thank the reviewer for taking the time to review our work and providing important suggestions. During the rebuttal, we have thoroughly addressed the concerns raised in the weakness, including:
> > > 1. Clarifying the connection of signals over groups and our method to the existing equivariant architecture, which is now illustrated in Sec A6.
> > >
> > > 2. Clarification on the use of interpolation operation (Clarified in Updated text Line 290)
> > >
> > > We also addressed the questions raised by the reviewer, including:
> > >
> > > 1. Clarifying what a "suitable subgroup" implies and how we verified our claim on selecting a suitable subgroup. We also clarified what is implied by 'task' in that context (Clarified in updated text, Line 58).
> > >
> > > 2. Clarification on the "Minimal Generating Set" that even if $e \notin S$, $S$ can generate the whole group $G$ .
> > >
> > > We thank the reviewer for pointing out typos. We have addressed them in the updated manuscript. Please review our responses and let us know if additional clarification or modification is needed.

---

> ### Comment · Reviewer_fTfZ · 2024-12-02
>
> I appreciate the author's rebuttal. I greatly appreciate the explanation regarding the connection with existing architectures, and I think Figure A.10 is a great visual. (You could also incorporate $\psi$ somehow to make it a bit more complete.) I have one small follow-up on that discussion: under this presentation, 5.1 should still have the form of $f: \mathbb{Z} \to \mathbb{R}$ with the lifting to $\mathbb{Z}\rtimes C_n$, which would still distinguish between the signal entries and the group, but Figure 3 still suggests they are intertwined.
>
> The introduction of the subsampling before the interpolation now makes sense to me, and I agree is reasonable and necessary for the section.
>
> Regardless of a response, I am updating my score slightly as I believe the main disagreement (lifting the group) could be reasonably explained and the subsampling comment was completely addressed.

---

> > ### Author Response · Authors · 2024-12-03
> >
> > We thank the reviewer for appreciating our work and taking the time to provide additional feedback.
> >
> > `Clarificaion on Sec 5.1`
> >
> > For ease of discussion, we explain using the rotation group $C_N$.
> >
> > We discuss in Sec. A6 that the image  $f: \mathbb{Z}^2 \rightarrow \mathbb{R}^k$ is lifted to $\mathcal{E}:\mathbb{Z}^2 \rtimes C_N \rightarrow \mathbb{R}$, which is represented as a tensor of size $H \times W \times |C_N|$. At each spatial loction $(i,j)$ of the tensor $\mathcal{E}$, we get a function over $C_N$, i.e., $E_{i,j}(d) = \mathcal{E}(h=i, w=j,d)~~\forall d \in C_N$.
> >
> > In more detail, the rotation group $C_N$ acts on $\mathcal{E}$ first by spatial rotation of all the channels (of dimension $H\times W$) and a cyclic shift of the channels along dimension $C_N$. That means the elements of $E_{i,j}$ undergoes a cyclic shift (regular representation of $C_N$).
> >
> > In Fig. 3 (Sec. 5.1), we visualize the anti-aliasing filters only along $E_{i,j}$, i.e., only along the $C_N$ dimension. The connection between the elements or the graph structure, shown in Fig. 3, represents the underlying Cayley graph, which captures how the elements of $E_{i,j}$ permute under the group action.
> >
> > Please let us know if additional clarification or modification is needed.

---

### Official Review · Reviewer_tNpv · 2024-11-05

**Soundness:** 4
**Presentation:** 3
**Contribution:** 3
**Rating:** 8
**Confidence:** 2

**Summary:**

The paper proposes a framework for the subsampling of data in group equivariant architecture. The framework explains how to make sure the downsampling operations are anti-aliased, leading to an improved performance. Their approach shows how to choose a sub-sampled group to make sure that the map stays equivariant and how to make sure that data is not aliased.

**Strengths:**

The paper uses signal processing and group mathematical formulation to formulate how to properly downsample in group equivariance architectures. I appreciate the level of delicacy in discussing the network should be designed to "properly" process data to make sure that information is not corrupted by aliasing in each sub-sampling layer. The paper is written clearly and is well-organized.

**Weaknesses:**

While the method is thorough, the authors are losing an opportunity to demonstrate the implications and significance of anti-aliasing in an experiment. Given that accuracy and results are only shown on benchmark datasets such as MNIST and CIFAR, whether this approach would benefit large-scale experiments is unknown. Here are suggestions and clarifying comments for improvements:

The paper lacks information on how their work differs from the prior works listed in related works. For example, how is the literature currently performing such a task? Is the literature completely missing such a subsampling framework for equivariance networks, hence the absence of comparison baselines? Is the key message of the paper that prior works don't do anti-aliasing? Or they do sub-sampling differently, which results in a non-equivariant map.

The experiments can be more thorough. They are limited in its current stage.

- Can the author focus on an application where the impact and benefits of anti-aliasing is better shown? Perhaps for an application where high-frequency information is of importance. An analysis on the artifacts and impacts of those due to naive sub-sampling can also help.
Results on MNIST in its current form are limited.


Minor

- I suggest not using bold phrases (such as line 44) in the paper.

- In Example 1, the middle label should be (b).

- I suggest not coloring the Lemma, Claim, etc., and following ICLR formatting instructions.

**Questions:**

- Is this the "first" work on looking into the subgroup sampling theorem of groups? Xu et al. 2021 discuss group equivariant subsampling. How this work differs from this prior work? Are there other works in the literature that address the discussed issue? If yes, please include these in the intro.

- It would be nice to show with a toy example why subsampling in groups is not an intuitive task of "every other sample" and what can go wrong. In this version of the paper, this intuition is not well-discussed.

- Prior to this paper how the literature is doing sub-sampling on groups? Could Table 1 include such a comparison where other approaches fail?

---

> ### Author Response · Authors · 2024-11-20
> **Rebuttal**
>
> We thank the reviewer for appreciating our work and providing valuable feedback.
>
> `Q1. How this work differs from related works on Subgroup subsampling`
>
> We appreciate the opportunity to clarify how our work differs from prior literature.
>
> We first clarify how our work differs from existing work [1,2] on subgroup subsampling:
>
> [1] Xu et al., Group equivariant subsampling, NeurIPS 2021
>
> [2] Cohen & Welling, Group equivariant convolutional networks, ICML 2016
>
> In the paper, we highlighted four main differences from prior work [1,2] on subgroup subsampling:
>
> **a)** [1,2] assume that the subgroup of the subsampling operation is given. (See line 44). We do not make such an assumption.
>
> **b)** As the subgroup is given, they do not consider any notion of selecting a subgroup or sampling rate. In contrast, we propose an approach to determine a subgroup given a desired subsampling rate. (See Line 061, the first bullet.)
>
> **c)** [1,2] did not analyze the effect of subsampling from a signal processing perspective. This work introduces the notion of band-limitedness and the conditions for which perfect reconstruction can be achieved. (See Lin 063, the second bullet.)
>
> **d)** [1,2] did not consider any form of anti-aliasing for their subsampling layer. While [1] did mention anti-aliasing in their related work, their approach actually does not address it. In fact, without a concrete notion of bandlimited-ness, it is not possible to discuss anti-aliasing. Following the bandlimited-ness introduced in our work, we further propose an anti-aliasing operation. (See Line 066, the third bullet point.)
>
> In hindsight, anti-aliasing for subsampling on a group seems quite intuitive. However, to the best of our knowledge, we are not aware of any existing work discussing anti-aliasing for group equivariant models.
>
>
>
> ---
>
> Next, we now discuss how our work differs from prior work [4,5,6] relating to the sampling theorem:
>
> [4] Chen et al., Discrete Signal Processing on Graphs: Sampling Theory, IEEE TSP 2015
>
> [5] Dodson et al., Groups and the sampling theorem, Sampling Theory in Signal and Image Processing, 2007
>
> [6] McEwen et al., A novel sampling theorem on the rotation group, IEEE Signal Processing Letters, 2015
>
> We believe that the closest works are the ones that extend the classic sampling theorem. This includes the studying of bandlimited-ness on graphs [4], or the study of sampling theorem on cyclic and abelian groups [5,6]. (See Line, 083 in related works). In this work, we propose a subgroup sampling theorem for general finite groups that guarantees the perfect reconstruction of the signal on the whole group from the subsampled signal on the subgroup. In Example 3, we also demonstrated that the existing Sampling theorem for the cyclic group is a special case of our proposed theorem.
>
> ---
> To summarize, our contribution bridges this gap by extending classical signal processing concepts—such as Shannon's sampling theorem—to group-equivariant models. Specifically, we propose a novel sampling theorem for functions over groups and introduce an anti-aliasing operator that improves both accuracy and equivariance consistency without adding any new learning parameters or significant computational overhead. Also, we provide a subgroup subsampling algorithm, which returns the subgroups at a fixed rate. We also show that the existing concept of bandlimited-ness of the cyclic group is a special case of our proposed theorem (see Example 3).
>
> **The key message**: We extend the techniques from traditional signal processing to group equivariant architecture, establishing a connection between sampling rate and subgroup subsampling. We propose a generalization of the sampling theorem for functions over a group. This holds great theoretical significance. We show that, by considering proper signal processing analysis, we can design a downsampling layer to improve accuracy and consistency without introducing any additional parameters.
>
>
> ---
> **Regarding baseline comparisons:** In Tab 2, all columns where $\mathcal{P}_\mathcal{M*}$ is marked **X** is the baseline with the naive subsampling procedure [2] (i.e., without proper anti-aliasing).
>
> Also, in Appendix Sec A2.1, we discuss how the primary goals of [1] differ from ours and involve equivariant index selection, not considering the problems mentioned above, i.e., our work and [1] targets two very different problems. In Table  A1, we show that when our theory is applied to their approach, we achieve better results. Further, proof of the varsity and adaptability of our proposed technique.
>
> We believe this addresses the concern regarding comparison with previous works, baselines, and key ideas of our work. We appreciate and are open to any suggestions to further improve the work.

---

> > ### Author Response · Authors · 2024-11-20
> >
> > `Q2. More thorough and meaningful application in the experiments`
> >
> > Thanks for the feedback. We acknowledge that additional experiments could further validate our approach. As mentioned in the limitation section, we believe our contribution is mainly in the theoretical aspect; this is also reflected in our bullets of contributions. The paper aims to provide a theoretical foundation for signal processing on functions over groups, particularly in group-equivariant architectures in computer vision.
> >
> > The experiments on CIFAR-10 (natural images) and MNIST (handwritten digits) were chosen because they strike a balance between complexity and interpretability, allowing us to effectively demonstrate how our proposed theory improves both accuracy and equivariance in $G$-CNNs. For example, for MNIST, we know digits '6' and '9' are confounding classes under the group of 90-degree rotation.
> >
> > The natural images of CIFAR-10 serve as a more complex benchmark, composed of colored natural images, which contain more complex images with higher-frequency details compared to MNIST, which further validates our approach. These application / datasets have been widely used in previous works while validating newly developed theories related to group equivariant models [1,2,3,4].
> >
> > [1] Cohen & Welling, Group equivariant convolutional networks, ICML 2016
> >
> > [2] Weiler et al., Learning steerable filters for rotation equivariant CNNs, CVPR 2018
> >
> > [3] Weiler et al., General E(2) - Equivariant Steerable CNNs, NeurIPS 2019
> >
> > [4] Wang et al., A general theory of correct, incorrect, and extrinsic equivariance, NeurIPS 2023
> >
> >
> > In addition to demonstrating theoretical contributions, our experiments show that our approach improves accuracy and equivariance while reducing the parameter requirement by $50\%$ all while introducing no additional parameter of the anti-aliasing operator with minimal computation overhead.
> >
> > Our current works serve the purpose of filling the missing theoretical link between group theory and signal processing, which is of significant theoretical interest. The experiments are designed to empirically validate our theory alongside the formal proofs. In future work, we plan to extend our model to larger-scale image datasets which is beyond the scope of this work.
> >
> > We hope this clarifies our experimental choices and demonstrates that they are sufficient for validating our theoretical framework.
> >
> > ---
> > In response to experiments with more complex datasets with high-frequency details, we have conducted additional experiments on the **STL-10** [5] dataset with image size $96\times 96$ containing higher frequency details. We can observe that, in this case, our method also improves performance with reduced parameter count. Especially at a given sampling rate, when our proposed anti-aliasing is performed, we see significant improvement in $\tt ACC_{orbit}$ and $\tt ACC_{loc}$. See the tables below and the updated Appendix.
> >
> > [5] https://cs.stanford.edu/~acoates/stl10/

---

> > > ### Author Response · Authors · 2024-11-20
> > >
> > > ## Results on STL-10
> > >
> > > This experiment also verifies the reviewer's intuition that data with high-frequency content is more suitable to demonstrate our significance. We obtained better gains in STL-10 than in MNIST and CIFAR-10, as the image resolution in STL-10 was much higher with high-frequency details.
> > >
> > >
> > > ## SO(2) Symmerty
> > > | Initial Symmetry Group | Sampling Factor |$\mathcal{P}_{\mathcal{M*}}$| #params     | $\tt ACC_{no aug}$ | $\tt ACC_{orbit}$ | $\tt ACC_{loc}$ | $\mathcal{L}_{\tt equi}$ |
> > > |-------------------------|----------------|----------------------------|-------------|--------------------|-------------------|-----------------|--------------------------|
> > > | $C_{24}$               | -               |                            | 1.3M   | 0.54              | 0.30           | 0.34        | 0.16                 |
> > > | $C_{24}$               | $2$             |    $\checkmark$            |962K    | 0.60              | 0.37           | 0.42        | 0.16                 |
> > > | $C_{24}$               | $2$             |    X                       |962K    | 0.60              | 0.35           | 0.40        |0.17                  |
> > > | $C_{24}$               | $3$             |    $\checkmark$            |831K    | 0.62              | 0.37           | 0.42        | 0.16                 |
> > > | $C_{24}$               | $3$             |    X                       |831K    | 0.60              | 0.34           | 0.38        | 0.18                 |
> > >
> > >
> > >
> > > ## O(2) Symmetry
> > > | Initial Symmetry Group | Sampling Factor |$\mathcal{P}_{\mathcal{M*}}$| #params     | $\tt ACC_{no aug}$ | $\tt ACC_{orbit}$ | $\tt ACC_{loc}$ | $\mathcal{L}_{\tt equi}$ |
> > > |-------------------------|-----------------|---------------------------|-------------|--------------------|-------------------|-----------------|--------------------------|
> > > | $D_{24}$               | -               |                            | 1.3M        | 0.57              | 0.32            | 0.37        | 0.12                 |
> > > | $D_{24}$               | 2               |  $\checkmark$              | 962K        | 0.64              | 0.27           | 0.40         | 0.19                 |
> > > | $D_{24}$               | 2               |    X                       | 962K        | 0.61              | 0.26           | 0.40         | 0.20                 |
> > > | $D_{24}$               | 3               |   $\checkmark$             | 831K        | 0.64              | 0.39           | 0.44         | 0.17                 |
> > > | $D_{24}$               | 3               |    X                       | 831K        | 0.60              | 0.33           | 0.33         | 0.17                 |

---

> > > > ### Author Response · Authors · 2024-11-20
> > > >
> > > > `Q3. Suggestions on paper formatting and typos.`
> > > >
> > > >
> > > > Thanks for pointing out the typos and reformatting suggestions. We have updated the manuscript accordingly by removing bolded phrases and colors.
> > > >
> > > > `Q4. Relation to the work of Xu et al.`
> > > >
> > > > We appreciate the opportunity to clarify how our work differs from Xu et al. (2021). We provided a detailed comparison of our method with the work of Xu et al. (2021)[1] in Section A2.1.
> > > >
> > > > For convenience of discussion, here we are clarifying the difference again.
> > > >
> > > > The goal of [1] and our work is different.
> > > >
> > > > The main aim of [1] is to select indexes in a way that respects a specialized equivariance (see [1] lemma 2.1) that can work between groups and (pre-defined) subgroups. The work is closer to [2].
> > > >
> > > > [1] Xu et al., Group equivariant subsampling, NeurIPS 2021
> > > >
> > > > [2] Chaman & Dokmanic, Truly shift-invariant convolutional neural networks, CVPR 2021
> > > >
> > > >
> > > > On the other hand, we aim to achieve the following goals:
> > > >
> > > > i. Choose an appropriate subgroup to downsample, given a sampling rate to
> > > > ii. Avoid signal distortion with proper anti-aliasing while maintaining equivariance.
> > > >
> > > > The paper [1] does not address (i) or (ii). For (i), they assume that the *subgroup is given* apriori, i.e., no choice is needed. For (ii), they do not consider the effect of aliasing or study the notion of band-limitedness (i.e., they perform naive subsampling).
> > > >
> > > > Our work and the technique proposed in [1] are complementary approaches. The specialized index-picking mechanism can be used along with our method after we choose the subgroup following our algorithm and the anti-aliasing operation before subsampling.
> > > >
> > > > The result in Table A1 shows that our proposed anti-aliasing operator consistently reduces the equivariance error and improves accuracy when used with the equivariant index selection scheme of [1].
> > > >
> > > > Additionally, in Example 3, we show that the existing sampling theorem for cyclic groups is a special case of our theorem.
> > > >
> > > > `Q5. Example illustrating the difficulty in subgroup subsampling`
> > > >
> > > >
> > > > We have added an example in the updated paper. See Appendix Sec A1.4 and refer to this section when discussing the diffucity in subgroup subsampling (line 45).
> > > >
> > > > `Q6a. Prior to this paper how the literature is doing sub-sampling on groups?`
> > > >
> > > > To the best of our knowledge, there is no prior work that discussed the anti-aliasing operation for the general discrete finite groups.
> > > >
> > > > In Example 3, we show that the existing notion of bandlimitedness for cyclic groups is a special case of our proposed theorem. This highlights that our sampling theorem is more general, as previous results are special cases of our theorem.
> > > >
> > > > For more general discrete groups, existing models that perform subgroup sampling in equivariant CNNs do not consider the bandlimited-ness of the signal, i.e., they perform naive subsampling.
> > > >
> > > > `Q6b. On including prior Work on Subsampling in Table 1`
> > > >
> > > > We have updated Table 1 to include the reconstruction error when the signal is not bandlimited. As expected, we observe a significant increase in error.
> > > >
> > > > In Table 1, we empirically validate our proposed sampling theorem by demonstrating that we can exactly reconstruct the signal on the original larger group from the subsampled signal when it satisfies the conditions outlined in Claim 2. To the best of our knowledge, no existing work addresses bandlimitedness and anti-aliasing operators for functions on general discrete groups, which is a key contribution of our paper.

---

> > > > > ### Comment · Reviewer_tNpv · 2024-11-25
> > > > > **Reviewer's Response (New Visualization is Requested)**
> > > > >
> > > > > I thank the authors for the detailed rebuttal. Given the new discussion, it is now clear how their framework differs from the prior works.
> > > > >
> > > > > Following my comment on datasets where high frequencies are present or oscillatory frequencies are important: An additional visualization (qualitative analysis) on the reconstruction quality of their method and baselines (a study to show the propagation of artifacts within the network in the baseline and its improvement in the method) is a great addition. Looking forward to this.

---

> > > > > > ### Author Response · Authors · 2024-11-27
> > > > > > **Response to the comment of tNpv**
> > > > > >
> > > > > > Thanks for the suggestion of having additional visualization. We have now included visualizations on the propagation of artifacts within a network architecture without anti-aliasing (baseline) and with anti-aliasing (Ours). Please see the updated appendix Sec. A2.3 and Figures A1:A8. As can be seen, the error propagates deeper into the network much more when not using anti-aliasing.
> > > > > >
> > > > > > Please let us know if there are further suggestions or clarification needed. We greatly appreciate and thank the reviewer for improving this paper!

---

> > > > > > > ### Author Response · Authors · 2024-11-28
> > > > > > > **Response to the comment of tNpv**
> > > > > > >
> > > > > > > In Sec. A8, following the reviewer's suggestion, we have provided an additional visualization of the reconstruction quality of the latent feature. Fig. A11 shows that our proposed anti-aliasing operation enables us to reconstruct the feature on the whole group from the downsample feature on the subgroup. On the other hand, reconstruction without anti-aliasing does not accurately reconstruct the downsampled features.

---

> > > > > > > > ### Comment · Reviewer_tNpv · 2024-12-01
> > > > > > > >
> > > > > > > > I thank the authors for the additional visualizations. The new experiments added by the authors and the rebuttal address most of the concerns raised by reviewers. Overall, this paper has rigorous analysis and formulation on how to define and perform anti-aliasing subsampling in group equivariant networks. I find it useful for the community and recommend its acceptance to the conference. I have accordingly raised my score.
> > > > > > > >
> > > > > > > > I note that two reviewers haven't responded to the authors' rebuttal at the time I write this comment. I strongly suggest that they review and comment.
> > > > > > > >
> > > > > > > > minor: In Figure A8, please swap the rows to be consistent with other figures.

---

> > > > > > > > > ### Author Response · Authors · 2024-12-02
> > > > > > > > >
> > > > > > > > > We thank the reviewer for appreciating our work and taking the time to improve this work. We will modify Fig. A8 accordingly.
> > > > > > > > >
> > > > > > > > > Best,
> > > > > > > > >
> > > > > > > > > Authors

---

### Author Response · Authors · 2024-12-04
**General Response**

We sincerely thank all reviewers for their constructive feedback and for engaging in the discussion period. We appreciate the positive ratings and acknowledgment of our work's contributions.

Unfortunately, it seems that Reviewer **FQDU** was unable to reply to our most recent response. We have thoroughly addressed these concerns in our rebuttal by providing additional explanations and evidence. We look forward to receiving further feedback.

Finally, we thank the reviewers and the AC for the time and effort spent on the reviewing process.

---

### Meta-Review · Area_Chair_XjRA · 2024-12-26

**Metareview:**

The paper studies downsampling in group equivariant neural networks. The paper introduces a notion of downsampling by a factor R which is applicable to general finite groups. The idea is to form a subgroup G’ by replacing a generator s with its R power s^R, eliminating elements which are generated by s^1, s^2, \dots s^{R-1}. If R is a composite integer, this subsampling can be ``spread’’ over generators, replacing s_1 with s_1^{R_1}, s_2 with s_2^{R_2} and so forth, with R_1 R_2 … = R. The paper gives an algorithm for constructing the resulting subgroup based on the Cayley graph. The paper then develops a notion of bandlimiting and an ideal antialiasing for this downsampling operator. All of these constructions generalize standard downsampling over the cyclic group.

The main strength of the paper is its conceptual and theoretical contributions: it gives a principled method for performing downsampling (with appropriate antialiasing) in group CNN. This is important conceptually, since ad-hoc downsample (or downsampling without antialiasing) can break equivariance. The paper demonstrates how to algorithmically determine a subgroup to downsample to, and how to construct an appropriate antialiasing filter. The initial submission demonstrated that this helps to improve equivariance in certain settings (MNIST, CFAR-10). The revision augments these results with experiments on higher resolution data (STL10).

**Additional Comments On Reviewer Discussion:**

Reviewers raised a number of questions about the paper. Key points are listed below:
- [A] Relationship between the proposed approach and existing work both in group CNN and in signal processing [all reviewers]
- [B] Properties of group CNN (i.e., why feature maps are defined over groups rather than over some other index set) [fTfZ, FQDU]
- [C] Strength and scope of experiments [fTfZ, rHa3]

Issue [B] was well addressed in the author response and changes to the paper. As mentioned by the response, the standard approach to applying group CNN to data that is not defined over a group is to lift the signal to the group and then apply the CNN.

Regarding issue [C], reviewers found the additional experiments performed during the discussion period convincing — in particular experiments on higher resolution data (STL10).

---

### Decision · Program_Chairs · 2025-01-22

Accept (Poster)